# Evaluating Yangtze River Delta Urban Agglomeration flood risk using hybrid method of AutoML and AHP

Yu Gao [1,2], Haipeng Lu [1,2], Yaru Zhang[1,2], Hengxu Jin[1,2], Shuai Wu[3], Yixuan Gao[1,2], Shuliang Zhang[1,2*]

1 Key Laboratory of VGE of Ministry of Education, Nanjing Normal University, Nanjing 210023, China

2 Jiangsu Center for Collaborative Innovation in Geographical Information Resource Development and Application, Nanjing 210023, China

3 Lianyungang Real Estate Registry, Lianyungang 222006, China

*Correspondence to*: Shuliang Zhang (zhangshuliang@njnu.edu.cn)

**Abstract.** With rapid urbanization, the scientific assessment of disaster risk caused by flooding events has become an essential task for disaster prevention and mitigation. However, adaptively selecting optimal machine learning (ML) models for flood risk assessment and further conducting spatial and temporal analyses of flood risk characteristics in urban agglomerations remain challenging. This study, establishes a "H–E–V–R" risk assessment index system that integrates hazard, exposure, vulnerability, and resilience based on the factors influencing flood risk in the Yangtze River Delta Urban Agglomeration (YRDUA). Utilizing Automated Machine Learning (AutoML) and the Analytic Hierarchy Process (AHP), a comprehensive flood risk assessment model is constructed. Results indicate that, among those of different assessment models, the accuracy, precision, F1-score, and kappa coefficient of the Categorical Boosting (CatBoost) model for flooded point identification are the highest. Among the flood hazard factors, elevation ranks highest in importance, with a contribution rate of up to 68.55%. The spatial distribution of flood risk in the study area from 1990 to 2020 is heterogeneous, with an overall increasing risk trend. This study is of great significance for advancing disaster prevention, mitigation, and sustainable development in the YRDUA.

## 1 Introduction

Under global climate change and accelerated urbanization, China has been experiencing pervasive flooding ever more frequently (Tang et al., 2024). Floods threaten people's lives, hinder social development and cause huge economic losses in China (Anon, 2021; Echendu, 2020; Milanesi et al., 2015). Flood formation has been exacerbated by climate change and urbanization, leading to increased frequency, extent, and intensity of urban flooding, and impacting urban flood risk (Mahmoud and Gan, 2018; Khadka et al., 2023; Scott et al., 2023; Seemuangngam and Lin, 2024). Modern human society is faced with the possibility of serious flood hazards and associated challenges, and in addition to post-disaster emergency management, the scientific assessment of disaster risks arising from flood events has gradually become a crucial aspect in preventing and mitigating disasters.

Currently, most research in the field of flooding focuses on the flood risks of individual cities (Wang et
al., 2021, 2023b; Guan et al., 2024). However, in recent years, the frequency and intensity of urban
flooding in China have increased dramatically, and individual cities are no longer able to independently
mitigate the risks arising from floods. Studies indicate that China's flood risk management needs to be
transformed from the scale of isolated individual cities to the scale of urban agglomerations, conducted
in a regionally coordinated manner (Morales-Torres et al., 2016; Wang et al., 2023a). City clusters,
constituting the spatial organizational structure of cities that have reached an advanced stage of
development, have become key areas for regional disaster management and sustainable development.
Due to the unique geographical location and climate conditions of the YRDUA, as well as the impact of
urbanization over the past 30 years, the frequency and intensity of flood disasters have been increasing,
posing a serious threat to the sustainable development of cities. Therefore, implementing relevant
emergency management strategies for flood risks is urgently needed. Furthermore, the region comprises
multiple cities, among which distinct resource interactions, such as population mobility and risk transfer,
exist (Lu et al., 2022). Thus, it is essential to assess both the overall flood risk characteristics and changes
in the urban agglomeration, as well as the spatial correlations of flood risks between cities, explore the
mutual influences and interaction mechanisms among regional disaster risks, and provide a scientific
basis for sustainable development within the urban agglomeration (Xu et al., 2024).
Statistical analyses of historical disaster statistics (Lang et al., 2004), indicator systems methods (Wang
et al., 2018b), scenario simulations methods (Yang et al., 2018), and data-driven methods (Abu-Salih et
al., 2023), are the primary flood risk assessment method currently. With the development of artificial
intelligence technology, data-driven methods, such as machine learning, deep learning, and artificial
neural networks, have emerged, providing new opportunities for improving traditional flood risk
assessment methods (Liu Jiafu and Zhang Bai, 2015). Ensemble methods are a class of machine learning
(ML) techniques that combine multiple base learners to form a stronger predictive model (Webb and
Zheng, 2004). They are designed to overcome several limitations of individual models, such as high
variance, overfitting, sensitivity to noise, and poor generalization (Yang et al., 2013). By aggregating the
outputs of weak learners, ensemble methods significantly enhance model stability, accuracy, and
robustness—especially in high-dimensional and complex classification or regression tasks (Kazienko
et al., 2015). Various ensemble ML techniques, including bagging (e.g., Random Forest), boosting (e.g.,

XGBoost, CatBoost), and stacking, have been widely used in hydrology, with boosting algorithms in particular showing strong performance in flood prediction and risk assessment (Shafizadeh-Moghadam et al., 2018; Mirzaei et al., 2021; Yan et al., 2024). However, ensemble ML techniques often lack preprocessing and feature selection capabilities, and their application effects vary considerably across different regions. To fully mine data and discover more effective features, experts have proposed other solutions, namely hybrid models such as ANFIS, LSTM-ALO, and LSSVM-GSA (Nayak et al., 2004; Yuan et al., 2018; Adnan et al., 2017). These methods have achieved good performances for given hydrological time series, focusing more on data preprocessing and feature selection. Although research on data-driven urban flood risk assessment methods has increased, certain limitations remain. For example, the physical importance of urban hydrological processes is often ignored in the model assessment process, interpretation of the assessment results is weak, and quantifying the boundaries and scales is challenging (Abu-Salih et al., 2023; Guo et al., 2022).

Furthermore, attempting to combine the data processing and feature selection capabilities of hybrid models with those of ensemble models remains challenging (Li et al., 2017). While ML algorithms have demonstrated strong performance in many domains, no single algorithm consistently performs best across all types of problems (Wolpert and Macready, 1997). Therefore, to achieve optimal performance, it is essential to carefully configure key components of the ML pipeline, including feature engineering, model selection, and hyperparameter tuning (Li et al., 2017; Raschka, 2020). Hence, ML applications require the participation of many experts, leading to disproportionate costs for ML development and improvement (Wagenaar et al., 2020; Sarro et al., 2022; Rashidi Shikhteymour et al., 2023). The effectiveness of ML improves with experience, where "experience" refers to the model's iterative exposure to training data and its ability to learn patterns from labeled examples (Jordan and Mitchell, 2015; Nagarajah and Poravi, 2019). One key challenge addressed in this study is how to automatically optimize model components such as feature selection and algorithm configuration in flood risk prediction, while maintaining high accuracy and adaptability across complex hydrological conditions. AutoML is an innovative ML framework that automates key stages of the model development pipeline, including feature selection, model selection, hyperparameter tuning, and ensemble learning (He et al., 2021a). By addressing these challenges, AutoML reduces reliance on expert knowledge and minimizes subjectivity in model building (He et al., 2021b; Consuegra-Ayala et al., 2022). In the context of this study, AutoML

enables the automatic optimization of hazard factor selection, model construction, and parameter
adjustment for flood risk assessment tasks, thereby improving efficiency, objectivity, and reproducibility
in model development. However, AutoML has not been widely applied in the fields of hydrology and
disaster risk management, and research has mainly focused on optimizing the ensemble model to achieve
better performance (Özdemir et al., 2023). Continuous research has highlighted the potential role of
AutoML in flood risk detection and assessment (Guo et al., 2022; Vincent et al., 2023; Munim et al.,
2024). Guo et al. (2022) compared AutoML with three single ML algorithms (CatBoost, XGBoost, and
BPDNN) and concluded that AutoML performed better in building rapid warning and comprehensive
analysis models for urban waterlogging. The model based on AutoML can be applied to areas without
water level monitoring and achieve accurate predictions and rapid warnings of waterlogging depth (Guo
et al., 2022; Yan et al., 2024). Abu-Salih et al. (2023) proposed a data-driven flood risk area detection
model that combined the ensemble model with the AutoML tool and successfully solved the problems
of data balance and strategy modeling, while reducing the complexity of flood risk area prediction.
Previous studies have provided a theoretical basis and scientific reference for the application of AutoML
methods to flood risk assessment. However, the use of AutoML for research purposes is a complex issue,
and many new opportunities and challenges remain regarding its specific applications.
In the field of flood risk assessment, AutoML has been preliminarily demonstrated to perform well in
flood hazard prediction (Guo et al., 2022c). As an efficient "black-box" modeling approach, AutoML
provides strong support for flood risk modeling through automated feature selection, model training, and
parameter optimization (Hutter et al., 2019; He et al., 2021a). In urban agglomerations, flood risk
assessment is a highly complex task involving diverse natural and socioeconomic factors derived from
heterogeneous and often multi-source datasets (Wang et al., 2023c). These factors—such as rainfall,
topography, land use, drainage, and population density—differ in type and often interact in non-linear
and uncertain ways (Shuster et al., 2005; Zhang et al., 2017; Wang et al., 2018a). Under such complex
circumstances, AutoML struggles to systematically evaluate the multi-dimensional indicators of flood
risk. To address this limitation, this study introduces a multicriteria decision analysis (MCDA) approach
to quantify the importance of various indicators within the evaluation framework (Pham et al., 2021).
MCDA facilitates the integration of such heterogeneous indicators into a unified evaluation framework
by constructing structured weighting schemes, thereby aligning the assessment results more closely with
real-world conditions and expert knowledge (Fernández and Lutz, 2010). In cases where data are limited

or certain indicators are difficult to quantify, MCDA methods allow for the incorporation of expert judgment through scoring systems and pairwise comparison matrices, enhancing the practical applicability and robustness of the model (Hites et al., 2006). The analytic hierarchy process (AHP) is one of the most popular MCDA techniques (Donegan et al., 1992). This technique emphasizes the importance of the subjective judgment of decision makers and the consistency of pairwise comparisons of standards in the decision-making process (Saaty, 1980). Recent studies have focused on integrated frameworks of ML models and MCDA technology for flood hazard assessment (Kanani-Sadat et al., 2019; Khosravi et al., 2019; Gudiyangada Nachappa et al., 2020; Mia et al., 2023). However, research focusing on using an integrated framework of AutoML and AHP techniques is still limited.

This study develops a flood risk assessment model for the YRDUA by analyzing the factors influencing flood risk and integrating AutoML and AHP methods. In this model, AutoML is employed to construct the flood hazard sub-model, using indicators that represent natural environmental drivers as input features. The hazard is modeled as a binary classification problem (i.e., whether flooding occurs), and the resulting feature importance rankings provide an objective basis for subsequent indicator weighting. Nevertheless, as a data-driven approach, AutoML alone cannot structurally interpret the relative influence of social and systemic factors within a multi-dimensional flood risk assessment framework. Therefore, this study incorporates the AHP to calculate the weights of flood exposure, vulnerability, and resilience in the YRDUA, based on expert knowledge and existing literature. A regional flood risk zoning map is then generated. A comparative analysis with observed inundation points data shows a strong spatial alignment between the distribution of flooded points and the high to medium-high risk zones, highlighting the reliability and applicability of the proposed model. The remainder of this paper is structured as follows: Section 2 describes the study area, data sources, and methodology; Section 3 presents the results and analysis; Section 4 discusses the findings and their implications; and Section 5 concludes the study with key insights and recommendations.

**2 Materials and methods**

In this section, the study area is briefly introduced (Section 2.1), and each individual component of the study is further discussed along with the basic geographic information, meteorology, social statistics, historical disaster data, and other fields involved in the study of urban agglomeration flood disasters and

their risks (Section 2.2). The framework of the flood risk assessment model is shown in **Figure 1**. The
factors influencing flood risk in the YRDUA are explored, and a flood risk assessment index system is
established (Section 2.3). The optimal model in AutoML is selected to calculate the importance of flood
hazard and hazard characteristic factors (Section 2.4), and the model is combined with AHP to determine
the weight of each risk indicator (Section 2.5). Ultimately, a flood risk assessment model based on
AutoML and AHP is constructed.

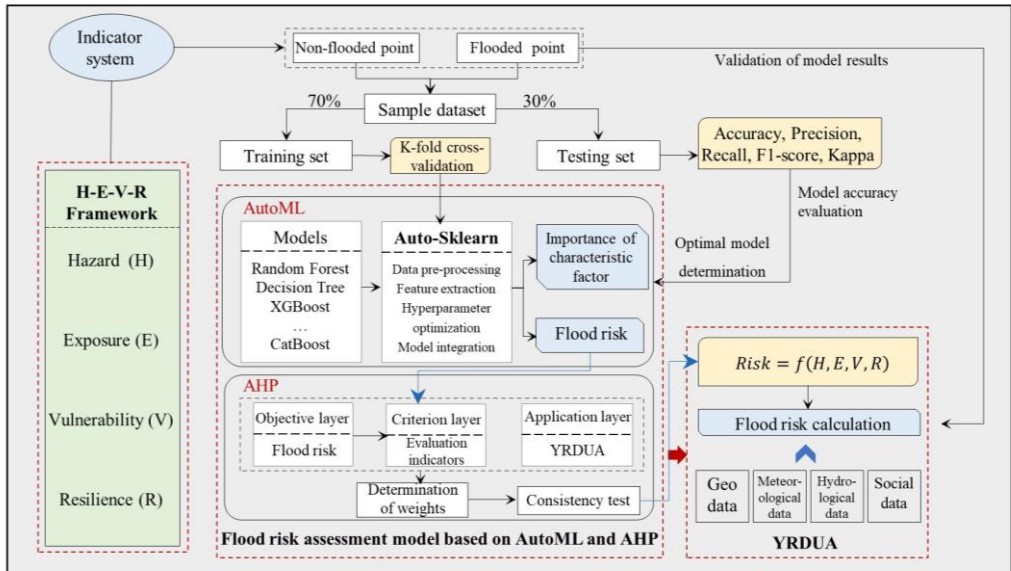


**Figure 1: Flood risk assessment modeling framework.**

**2.1 Study area**
The Yangtze River Delta Urban Agglomeration, located in the eastern coastal region of China (27° 04′–
34° 49′ N; 115° 75′–122° 95′ E), includes 27 cities: 8 in Anhui Province, 9 in Jiangsu Province, 9 in
Zhejiang Province, and Shanghai (**Figure 2**) (Yang et al., 2024). Influenced by the East Asian summer
monsoon, the study area features low-lying plains in the northern region and higher hilly terrain in the
southern region, along with numerous waterways (Ding et al., 2021). With the recent accelerated climate
change and urbanization, extreme precipitation events in the Yangtze River Delta (YRD) have been
occurring ever more frequently, and the temporal and spatial distribution differences in precipitation have
increased. Additionally, the increase in impervious surfaces, narrow plains rivers, and poor drainage may
result in more frequent and widespread urban flooding and waterlogging disasters (Wan et al., 2013).
This region is economically developed and densely populated , making it the largest urban agglomeration
in Asia (Sun et al., 2023). In 2008, the Gross Domestic Product (GDP) of the YRD accounted for 17.5%
of the GDP of the entire country, i.e., 4.3 trillion yuan, and the per capita GDP was 44,468 yuan, i.e.,
twice the national average level. The population has reached 97.2 million, i.e., 7.3% of China's total
population, and the region's average population density is 877 persons/km$^2$, i.e., approximately twice the
national average (Gu et al., 2011). Therefore, the potential risks of flood and waterlogging disasters are
substantial.

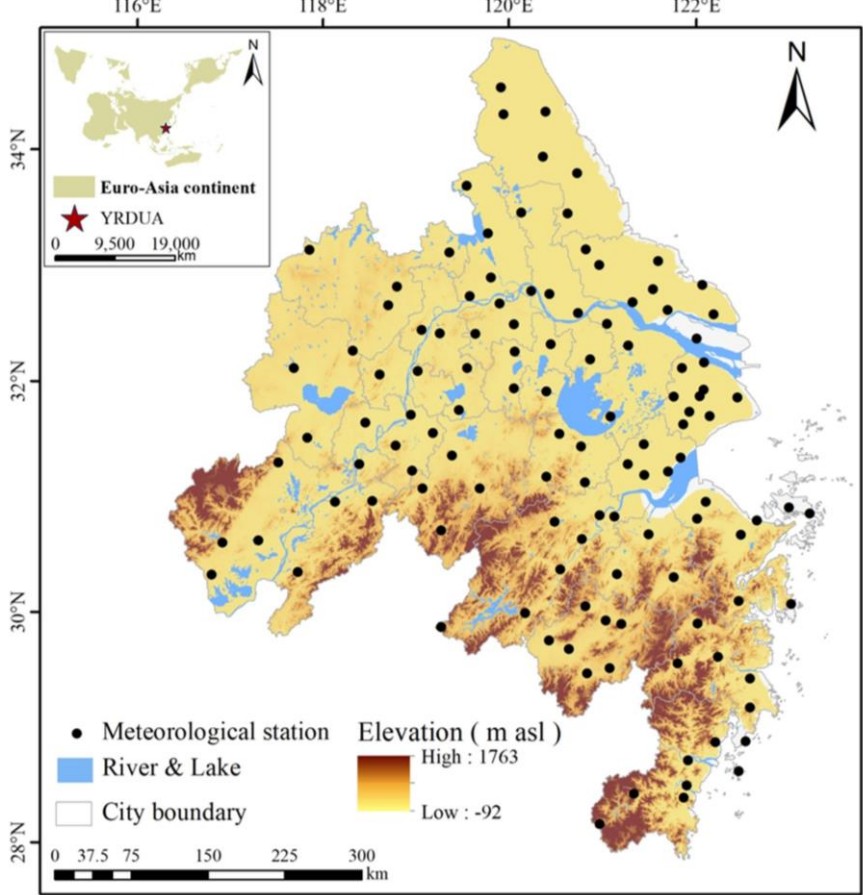

**Figure 2: The schematic map of the YRDUA.**
**2.2 Data sources and Processing**
**2.2.1 Data sources**
The study of flood disasters and their associated risks in urban agglomerations involves complex natural
and socio-economic factors. Therefore, we collected and preprocessed data from multiple fields, such as
basic geography, meteorology, social statistics, and historical disasters. **Table 1** lists the data types and
resolutions collected for the research area.
**Table 1: Description of the Datasets Used for Flood Risk Assessment, Their Characteristics, and Data Sources.**

| Category | Details | Resolution | Data Source |
|---|---|---|---|
| Basic Geographic Information Data | Administrative boundaries and river network density data. | 30m | -Resources and Environmental Science and Data Center, CAS (https://www.resdc.cn/). -USGS (https://earthexplorer.usgs.gov/). -Wuhan University CLCD dataset (https://zenodo.org/records/8176941). - National Ecosystem Science Data Center (nesdc.org.cn). |
| | Digital Elevation Model (DEM) based on SRTM1 (30m), mosaicked and clipped to the study area (27 core cities). | | |
| | Land use data from CLCD (30m), includes 7 types: farmland, forest, shrubland, grassland, water, bare land, and impervious surfaces. | | |
| | NDVI data (2000–2020) calculated using the GEE platform. | | |
| Meteorological Data | Hourly precipitation data from 120 meteorological stations. Data preprocessed for outlier removal and missing value handling. | Station data | National Meteorological Information Center, China Meteorological Administration |
| Social Statistics | Population, unemployment, GDP, and healthcare statistics at the prefecture level. | Prefecture-level | Provincial and municipal statistical yearbooks and bulletins |
| | Urbanization rate calculated using urban population proportion. | | |
| | GDP density and per capita GDP derived from total GDP and land area/population. | | |
| Historical Disaster Data | Flood inundation data from the MODIS-based Global Flood Database (2000–2018), processed to focus on the YRDUA region. To ensure comprehensive selection of inundation points, the inundated areas within the time frame were overlaid to produce a historical flood map. | 250m | Global Flood Database (https://www.emdat.be/). |


**2.2.2 Data Standardization and Preprocessing**
Due to variations in data sources and formats, the collected flood disaster risk data exhibit differences in
spatial resolution, dimensions, and magnitude. To ensure consistency and comparability, standardization
of both spatial scale and numerical range was performed before using these datasets as flood risk
indicators.
(1) Unification of spatial scale means aligning data within the same coordinate range and resolution. The
research data is standardized through projection transformation, converting all datasets into the same
geographic and projected coordinate systems. To generate continuous spatial surfaces from discrete data
points, we applied the Ordinary Kriging interpolation method, which assumes a constant but unknown
local mean (Cressie, 1990). A spherical semivariogram model was adopted to capture spatial
autocorrelation, as it is widely used in environmental geostatistics for its bounded range and smooth
continuity (Webster and Oliver, 2007). The interpolation process was carried out using ArcGIS 10.8.
Finally, if the spatial data has different resolutions, resampling is performed to standardize all data to the
same resolution, which in this study is unified to 30m×30m.
(2) Normalization of the numerical range can be achieved using a normalization process. In this study,
the Min-Max normalization method is applied. Specifically, the minimum and maximum values of each
feature are computed only from the training set, and both the training and test sets are then normalized
using these training-derived parameters. This ensures that the normalized values in the training set are
scaled to the range [0,1], while the values in the test set may exceed this range if they fall outside the
training set's value distribution. The formula is as follows:
$$x' = \frac{x - x_{min}^{train}}{x_{max}^{train} - x_{min}^{train}} \tag{1}$$
**2.3 Historical Flood Inundation Point Extraction**
The historical flood inundation map of the study area is shown in **Figure 3** (a). The flood inventory map
used in this study was created based on inundation data from the Global Flood Database and the EM-
DAT flood disaster database, and further verified through satellite imagery, Google Earth, and
documented historical flood records. The actual flooded areas were delineated from flood traces in the
inundation dataset and image interpretation. A flooded point is defined as a location that lies within the
inundation extent of at least one recorded flood event during the study period. Based on this definition,

278 flooded points were randomly selected from the validated inundated areas. These points serve as the foundation for subsequent statistical analysis and model training, with their spatial distribution shown in **Figure 3** (b).

To calculate flood hazard, it is necessary to select training samples. The task of identifying flooded and non-flooded points using AutoML is essentially a binary classification problem, which requires a balanced number of samples. An imbalanced ratio of positive and negative samples can result in unreliable classification outcomes. Previous studies (Pham et al., 2021; Bostan et al., 2012) have shown that the best classification performance is achieved when the ratio of flooded to non-flooded points is 1:1. Therefore, after selecting the flooded points, 278 non-flooded points were randomly sampled to ensure a balanced 1:1 ratio, excluding high-altitude areas, based on the region's actual characteristics. Finally, the flooded and non-flooded points were used as sample data and divided into a 7:3 ratio (70% for training and 30% for testing) for model training.

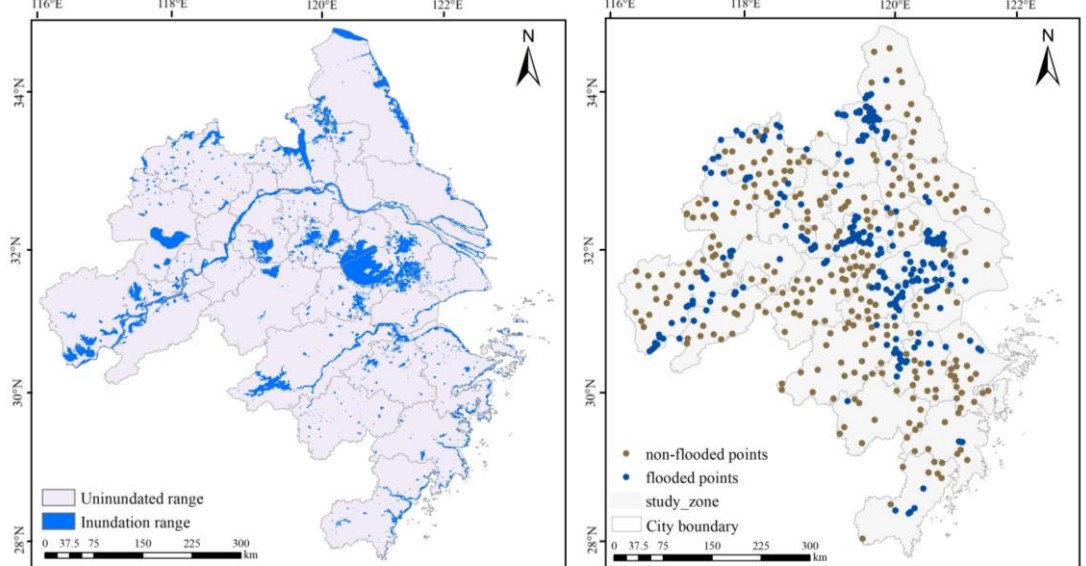

**Figure 3: (a) Flood inundation map of the study area. (b) Spatial Distribution of Flooded and Non-Flooded Points in the YRDUA.**

**2.4 Establishment of a flood risk assessment indicator system**

Although risk is a universal concept, it has no universal definition (Aven, 2016; Mishra and Sinha, 2020). Based on the hazard–exposure–vulnerability (H–E–V) disaster risk framework, we considered the particularity of flood risk research at the urban agglomeration scale, incorporated resilience indicators

into the existing framework, and constructed a four-dimensional flood risk assessment framework of
hazard–exposure–vulnerability–resilience (H–E–V–R) that can assess regional flood risks more
comprehensively and systematically. The conceptual description of flood risk in this study can be
expressed in the Eq. (2):
$Risk = f(H, E, V, R) = \sum_{i=1}^{a} \omega_H H_i + \sum_{i=1}^{b} \omega_E E_i + \sum_{i=1}^{c} \omega_V V_i + \sum_{i=1}^{d} \omega_R R_i,$ (2)
where $H$, $E$, $V$, and $R$ represent the danger of, exposure to, vulnerability to, and resilience in response
to floods, respectively; $\omega_H$, $\omega_E$, $\omega_V$, and $\omega_R$ are the weights of danger, exposure, vulnerability, and
resilience, respectively; $H_i$, $E_i$, $V_i$, and $R_i$ are the values of items $i$ of the indicators, respectively; and
$a$, $b$, $c$, and $d$ are the numbers of the indicators, respectively.
We constructed a flood risk assessment index system for the YRDUA based on the "H–E–V–R"
framework, the actual situation of the study area, the formation mechanisms of flood disasters, and the
findings of relevant studies (Gain et al., 2015; Criado et al., 2019; Hsiao et al., 2021). We selected four
first-level indicators (i.e., hazard, exposure, vulnerability, and resilience indices) and 19 second-level
indicators.  A detailed description of the flood risk assessment index system is presented in **Figure 4**.

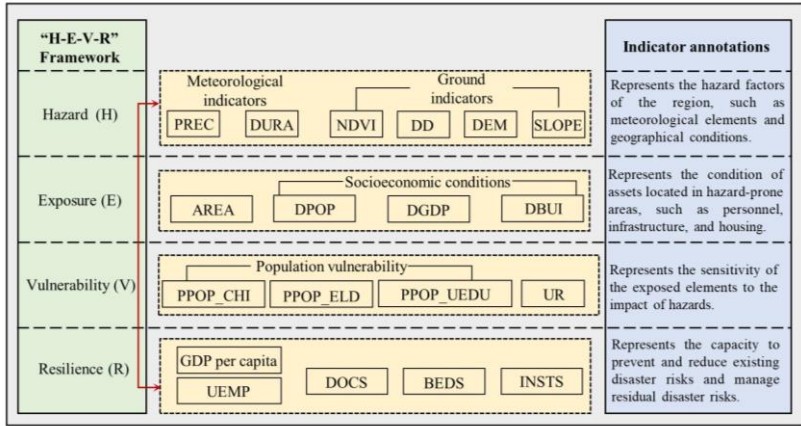

**Figure 4: Flood risk assessment index system for the YRDUA based on the H–E–V–R framework.**

The hazard indicators consisted of six indices: Average annual precipitation (PREC), Annual Cumulative
Heavy Rainfall Duration (DURA), Digital Elevation Model (DEM), SLOPE, Drainage Density (DD),
and Normalized Difference Vegetation Index (NDVI). Rainfall is the primary factor leading to flooding,
particularly extreme rainstorms caused by climate change. According to the Meteorological Bureau's
definition, a heavy rainstorm event is characterized by rainfall of 50mm or more within 24 hours. DURA
is defined as the total number of days with heavy rainstorm events occurring at all meteorological stations
within the study area each year. The more days heavy rainstorms accumulate and the longer their duration,
the greater the likelihood of flooding and other disaster events. DEM and SLOPE are important
topographical indicators. Areas with low DEM and SLOPE values are generally more susceptible to
flood threats. DD refers to the area of rivers or lakes per unit of land surface area and is a crucial indicator
of a watershed's structural characteristics. It determines the watershed's capacity to withstand flooding.
The higher the DD, the greater the likelihood of flooding and the higher the potential flood risk.
Vegetation plays a role in water storage, retention, and infiltration. The lower the vegetation coverage,
the weaker the buffering capacity, making it more likely for surface water to accumulate. The NDVI
index measures the relative abundance of green vegetation, with values ranging from -1 to 1. The higher
the value, the greater the vegetation coverage, and the lower the risk of flooding.
Land area (AREA), Population Density (DPOP), GDP Density (DGDP), and Building Density (DBUI)
were selected as exposure indicators to assess the degree of vulnerability of both the natural environment
and social systems to flooding. The land area for each administrative unit at the prefecture-level city is
calculated individually. A larger land area corresponds to a greater extent exposed to flooding. DPOP
and DGDP represent the concentration of population and assets, respectively. Areas with higher DPOP
and DGDP are more susceptible to potential threats from pluvial flooding. DBUI, the ratio of total
building area to total land area in a region, indicates the building density. A higher DBUI reflects greater
exposure to flooding.
Vulnerability indicators focus more on the social aspects of flood disasters. This study selects four
vulnerability indicators: Proportion of Child Population (PPOP_CHI), Proportion of Elderly Population
(PPOP_ELD), Proportion of Uneducated Population (PPOP_UEDU), and Urbanization Rate (UR). Age
is a key feature of social vulnerability, and both the population aged 0-14 and those over 65 are considered
vulnerable groups, as these age groups are more susceptible to flood damage. The uneducated population
generally has a weaker awareness of disaster risks and lower self-protection capacity, which makes this
group more vulnerable to flooding. The urbanization rate refers to the proportion of the urban population
in the total resident population of a region. This indicator is inversely related to flood vulnerability. In
general, a higher urbanization rate indicates greater social development and stronger protective capacities,
which can reduce vulnerability to flooding to some extent.
The resilience indicators selected in this study include Gross Domestic Product (GDP) per capita,
Unemployment Rate (UEMP), Number of Doctors (DOCS), Number of Medical Institutions (INSTS),
and Number of Hospital Beds (BEDS). GDP per capita is the ratio of a region's GDP to its total resident
population over a specified period, reflecting the region's economic condition. A higher GDP per capita
indicates a more developed economy, which is associated with a greater capacity to recover quickly after
a flooding event. The Unemployment Rate (UEMP) measures the proportion of the idle labor force,
indirectly reflecting the stability of urban development. A high unemployment rate signals economic
instability, which weakens the capacity to cope with floods and extends the time required for post-disaster
recovery, thus impeding disaster response efforts. The indicators of DOCS, INSTS, and BEDS provide
insights into a region's healthcare and medical support capabilities. Areas with stronger healthcare
systems are better positioned to manage flood risks and recover more effectively from such disasters.

**2.5 Flood risk calculation method based on AutoML**
**2.5.1 Feature selection**
The flood inventory map in this paper was developed using inundation data from the Global Flood
Database and flood disaster data from the EM-DAT database, supplemented by satellite and Google
image interpretation and verified against existing historical flood records. The actual flood-affected areas
were delineated based on flood traces from the inundation datasets and image interpretations. For this
study, 278 flood inundation points were randomly selected within the inundation data range during the
study period, and the location of each point was used as the basis for subsequent statistical analysis of
flood events. The main factors affecting flood risk were considered during input feature selection.
Rainfall and rainstorms are important factors that lead to floods, and flooding is closely related to
topography, slope, vegetation cover, and hydrological conditions. Therefore, six indicator factors,
namely PREC, DURA, DEM, SLOPE, NDVI, and DD, were selected as the input features of the model.
To verify the model, 70% of the data in the sample were set as the training dataset and the remaining 30%
of the data were set as the testing dataset through random sampling.
When the number of samples is small, data balancing is essential to ensure uniform sampling and reduce
the deviations among the training, validation, and original datasets. Data balancing refers to the process
of achieving a balanced distribution of data for each labeled category; it is particularly important when
the number of observations in each class is significantly different. One way to address an imbalanced
dataset is to oversample the minority classes. In this study, we assessed flood risk based on the
identification of flooded point in the sample, which is essentially a binary classification problem;
therefore, the output features are 0, i.e., negative categories (non-flooded points), versus 1, i.e., positive
categories (flooded points). The processed dataset comprised 278 positive samples (flooded points) and
278 negative samples (non-flooded points), with each label consisting of 278 points representing the
entire dataset.
**2.5.2 Model training and hyperparameter optimization**
Training samples were generated using the data from flooded and non-flooded points in the study area,
and the Auto-Sklearn was used for model training, its principle is shown in **Figure 5**. The Auto-sklearn
framework has multiple built-in machine learning algorithms. We selected 9 models that are more typical
or have better performance in flood hazard research: random forest (RF), extreme gradient boosting
(XGBoost), Light Gradient Boosting Machine (LightGBM), categorical feature boosting (CatBoost),
Extra trees, Decision tree, Nearest Neighbors, Multi-layer Perceptron (MLP) neural network, and Linear
Regression. The training and testing datasets were used to train the 9 machine learning models, and the
hyperparameters were continuously adjusted and optimized.

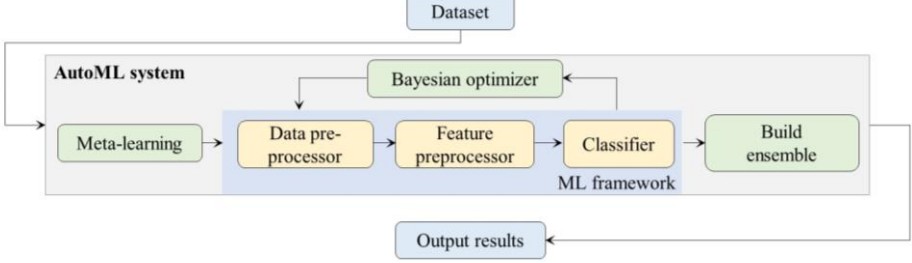


**Figure 5: Principles of Auto-Sklearn.**

Hyperparameter optimization is an important step in ML model training. The aim of this step is to
determine a hyperparameter combination to generate a ML model that performs well on a specific dataset
and reduces the effect of the predefined loss function on a given dataset. In this study, we used a grid
search strategy for optimization. For each set of hyperparameter combinations, k-fold cross-validation
was used to evaluate the model. To quantify the balance between Precision and Recall, the F1-score was
used as the primary evaluation metric. The hyperparameter combination corresponding to the model with
the highest average F1-score was selected as optimal. Briefly, the training dataset was divided into K
parts, of which one was selected as the test set and the rest were used as the training set. The cross-
validation was repeated K times and the results were averaged K times. The model with the best average
result among all models was selected as the optimal model, and the final classification prediction result
was the output. In this study, we used 5-fold cross-validation.
It is important to note that 5-fold cross-validation was employed at two distinct stages in this study. First,
it was conducted within the training set during hyperparameter tuning as part of the AutoML model
selection process. Second, following final model selection, an independent 5-fold cross-validation was
applied to the entire dataset to evaluate the generalization performance of the model and identify potential
overfitting. The data partitions used in the two stages were entirely separate, ensuring that no data leakage
occurred.
**2.5.3 Performance evaluation**
To better compare the performance of the 9 selected ML models in the Auto-Sklearn framework for flood
risk assessment, multiple evaluation indicators were used to assess the test dataset. The following
combinations of the true category of the sample point and the category predicted by the classifier were
used: True Positive (TP)—the sample point is a flooded point, and the model classifier also predicts that
it is a flooded point; True Negative (TN)—the sample point is a non-flooded point, and the model
classifier also predicts that it is a non-flooded point; False Positive (FP)—the sample point is a non-
flooded point, and the model classifier mistakenly predicts that it is a flooded point; False Negative
(FN)—the sample point is a flooded point, and the model classifier mistakenly predicts that it is a non-
flooded point. Therefore, four related indicators were selected: Precision, Recall and F1-score, and the
consistency metric Kappa coefficient. The calculation formulas are as follows:
$Precision = \frac{TP}{TP+FP}$ ,                                                           (3)
$Recall = \frac{TP}{TP+FN}$ ,                                                             (4)
$F1-_{score} = \frac{2TP}{2TP+FP+FN}$ ,                                                  (5)
Among the indicators, *Precision* refers to the proportion of correctly predicted flooded points among all
predicted flooded points, reflecting the model's ability to avoid false positives. *Recall* measures the
proportion of correctly identified flooded points among all actual flooded points, representing the
model's sensitivity. *F1-score* is the harmonic mean of Precision and Recall, providing a balanced
evaluation of both metrics and reflecting the overall recognition performance of the model.
*Kappa* coefficient is a statistical consistency metric used to measure classification performance, which
is calculated based on the confusion matrix of true and predicted categories. Its value ranges from [-1,1]:
A Kappa value of 1 indicates perfect agreement, 0 means the classification is no better than random
guessing, and negative values suggest the classification is worse than random prediction. *Kappa* is
calculated using Eq. (6), where $P_e$ is given by Eq. (7).
$$Kappa = \frac{P_o - P_e}{1 - P_e}, \tag{6}$$
$$P_e = \frac{(TP+FP)(TP+FN)+(TN+FN)(TN+FP)}{(TP+FP+TN+FN)^2}, \tag{7}$$
where $P_e$ represents the expected agreement by chance.
Combining multiple indicators allows for a more comprehensive evaluation of models within the Auto-
Sklearn framework for flood point identification and flood risk assessment.
**2.6 Method for determining flood risk index weights based on AHP**
**2.6.1 Establishing a hierarchical model**
According to the decision-making objectives, factors, and applications in decision-making problems, the
AHP can be divided from bottom to top into the target, criterion, and application layers. Among them,
the target layer is the problem to be solved (i.e., final flood risk). The criterion layer is the intermediate
link, including the factors to be considered and the decision-making criteria. The factors can be divided
into different evaluation indicators, including four first-level indicators (danger, exposure, vulnerability,
and resilience) and their corresponding 19 second-level indicators. The criterion layer comprises various
weight combination schemes linked to the target layer. The application layer is the final optional scheme
and specific application of the decision. The final weight scheme and evaluation results of this study
were applied to the YRDUA.

## 2.6.2 Constructing the judgment matrix

After the hierarchical structure was established, a judgment matrix was constructed based on the relationship between the criteria and indicators. Different elements in the sublevel were compared pairwise, and the relative importances of all elements in the current layer and previous layer were compared. Typically, a pairwise comparison matrix is used as representative. In this study, we adopted the 1–9 scale method as the importance measurement standard. The importance comparison relationship is presented in **Table 2**, where the matrix element $a_{ij}$ represents the comparison result of the $ith$ element relative to the $jth$ element.

**Table 2: Pairwise comparison point-based rating scale of AHP.**

| Ranking | Importance Level |
|---------|------------------|
| 1 | Equally important |
| 3 | $i$ is slightly more important than $j$ |
| 5 | $i$ is much more important than $j$ |
| 7 | $i$ is very much more important than $j$ |
| 9 | $i$ is extremely important than $j$ |
| 2, 4, 6, 8 | Intermediate value of two adjacent judgments |
| Reciprocal | Comparative judgment of j vs., $a_{ji} = 1/a_{ij}$ |

## 2.6.3 Solving the eigenvector of the judgment matrix

Based on the judgment matrix, the square root method was used to solve the eigenvector and eigenroot. The first step is to calculate the square root $a_{ij}$ of the product of each row of the judgment matrix $n$, then normalize it, and finally calculate the maximum eigenroot of the judgment matrix. The formula is as Eq. (8), Eq. (9), Eq. (10).

$$M_i = \sqrt[n]{\prod_{j=1}^{n} a_{ij}}, \tag{8}$$

$$W_i = \frac{M_i}{\sum_{i=1}^{n} M_i}, \tag{9}$$

$$\lambda_{max} = \sum_{i=1}^{n} \frac{(AW)_i}{nW_i}, \tag{10}$$

**2.6.4 Consistency check**
After the eigenvector calculation is completed, a consistency test is required to reduce the subjectivity in
the judgment matrix and enhance the scientific nature of the data and calculations. In a pairwise
comparison matrix, consistency means that the decision-maker's judgments must exhibit logical
coherence and transitivity. Specifically, if option A is considered more important than option B, and
option B is considered more important than option C, then consistency requires that option A must also
be judged more important than option C (Saaty, 1984).
The consistency indicator (CI) is used to measure the deviation of the judgment matrix from the
consistency: the smaller the CI, the greater the consistency of the judgment matrix. When CI = 0, the
judgment matrix is completely consistent. The CI calculation formula is as Eq. (11).
$$CI = \frac{\lambda_{max} - n}{n - 1},$$ (11)
To quantify the standard, the relative consistency (CR) index was further calculated as Eq. (12).
$$CR = \frac{CI}{RI},$$ (12)
Where average Random Consistency Index (RI) represents the average random consistency which
depends only on the order of the judgment matrix. The RI values for judgment matrices of order 1 to 10
are shown in **Table 3**.
**Table 3: Consistency index (RI) for a randomly generated matrix.**

| n | 1 | 2 | 3 | 4 | 5 | 6 | 7 | 8 | 9 | 10 |
|---|---|---|---|---|---|---|---|---|---|---|
| RI | 0.00 | 0.00 | 0.52 | 0.89 | 1.12 | 1.26 | 1.36 | 1.41 | 1.46 | 1.49 |


CR was determined based on the RI value. When CR < 0.1, the consistency of the judgment matrix is
considered good. When CR > 0.1, the consistency of the judgment matrix is unacceptable, and the
judgment matrix must be adjusted and modified. In such cases, the corresponding judgment matrix was
further constructed, and the eigenvector and eigenroot were calculated using the following formulas:
Finally, the judgment matrix that passed the consistency test was used to calculate the weights of the
indicators at the different levels.
**2.7 Determination of flood risk levels**
The classification of flood risk levels often involves manually setting thresholds, which can introduce
subjectivity and influence the accuracy of the risk assessment outcomes (Ma et al., 2022). To calculate
the flood risk, we employed the natural breakpoint classification method, which groups data into classes
based on natural divisions within the dataset (Lin et al., 2019). This method works by identifying points
where the data distribution changes most significantly and dividing the data into ranges based on these
breaks. Unlike clustering methods, which do not focus on the number and range of elements in each
group, the natural breakpoint method ensures that the range and number of elements in each group are as
balanced as possible (Ma et al., 2022).
**3 Results and discussion**
**3.1 Model flood risk results and evaluation**
**3.1.1 AutoML optimal model selection**
In the experiment, 9 typical ML models under the Auto-Sklearn framework were used to process the
sample dataset, with 70% of the sample set being used as the training dataset and 30% being used as the
testing dataset. The results of the comparative analysis of the model performance based on the test dataset
are presented in **Table 4**. A comprehensive analysis of the results on the testing data revealed that, in
terms of Precision, CatBoost had the highest value (0.9030), followed by LightGBM (0.8960) and Extra
Trees (0.8893). Meanwhile, CatBoost had the highest recall rate of 0.8883, followed by that of Extra
Trees at 0.8870. The probability thresholds for Precision, Recall, and F1-score range from [0,1], while
the Kappa coefficient ranges from [-1,1]. The F1-score and Kappa coefficient of the CatBoost model
were also markedly higher than those of the other models, reflecting the model's good consistency. A
comprehensive comparison showed that the precision, F1-score, and Kappa coefficient of the CatBoost
model were the highest, with its precision reaching 0.9030, indicating that the recognition and prediction
precision of the flooded points in the study area based on the CatBoost model were obviously better than
those of other common machine learning models.
Since flood data often involve various environmental factors and complex interactions, the CatBoost
model is highly effective at handling these nonlinear relationships and feature interactions. Additionally,
the model incorporates multiple regularization mechanisms during tree construction, which helps reduce

overfitting and enhances the model's generalization capability. As shown in **Table 4**, most models performed well on the training set, but their performance slightly declined on the test set, highlighting variations in generalization ability. CatBoost demonstrated strong robustness, achieving a Precision of 0.9319 on the training set and 0.9030 on the test set. Additionally, LightGBM and XGBoost showed relatively consistent performance between the training and test sets, suggesting better generalization. However, models such as Decision Tree and Nearest Neighbors exhibited a more significant performance drop in the test set, indicating a higher sensitivity to overfitting. Interestingly, in a few cases (e.g., Extra Trees), test set performance slightly exceeded that of the training set in certain metrics. This is not uncommon in small, balanced datasets and may result from a combination of factors such as random sampling variation, slightly easier test samples, or appropriate regularization that reduces overfitting in the training set. To further evaluate overfitting, we used 5-fold cross-validation by comparing the performance of the training and testing sets. The experimental results indicate that while most models showed some performance decline on the test set, CatBoost maintained relatively stable performance, suggesting that the model does not exhibit significant overfitting and has good generalization ability.

**Table 4: Comparative analysis of the performances of different ML models.**

| Models | Dataset | Precision | Recall | F1-score | Kappa |
|---|---|---|---|---|---|
| CatBoost | Training set | 0.9319 | 0.9307 | 0.9547 | 0.8614 |
| | Testing set | 0.9030 | 0.8883 | 0.8960 | 0.7915 |
| XGBoost | Training set | 0.9017 | 0.8827 | 0.8818 | 0.8640 |
| | Testing set | 0.8748 | 0.8640 | 0.8624 | 0.7256 |
| LightGBM | Training set | 0.9349 | 0.9307 | 0.9306 | 0.8616 |
| | Testing set | 0.8960 | 0.7890 | 0.8015 | 0.7324 |
| Random Forest | Training set | 0.8922 | 0.8747 | 0.8745 | 0.7484 |
| | Testing set | 0.8482 | 0.8320 | 0.8309 | 0.6662 |
| Extra Trees | Training set | 0.8524 | 0.8240 | 0.8735 | 0.7695 |
| | Testing set | 0.8893 | 0.8570 | 0.8877 | 0.7751 |
| Decision Tree | Training set | 0.8886 | 0.8640 | 0.8621 | 0.8040 |
| | Testing set | 0.8810 | 0.8720 | 0.8708 | 0.7419 |
| Linear Regression | Training set | 0.8636 | 0.8533 | 0.8525 | 0.7073 |
| | Testing set | 0.8682 | 0.8480 | 0.8450 | 0.6926 |
| Nearest Neighbors | Training set | 0.7301 | 0.7907 | 0.7987 | 0.6009 |
| | Testing set | 0.7747 | 0.7440 | 0.7390 | 0.4937 |

| Models | Dataset | Precision | Recall | F1-score | Kappa |
|---|---|---|---|---|---|
| MLP Neural Network | Training set | 0.8998 | 0.8880 | 0.8873 | 0.7765 |
| | Testing set | 0.8682 | 0.8480 | 0.8450 | 0.6926 |

475

By comparing the performances of the 9 models, we found that the CatBoost model was more effective

in identifying flooded points. To further verify the excellent performance of the model, the receiver

operating characteristic (ROC) curve and area enclosed by the coordinate axes (corresponding area under

the curve [AUC] value) were plotted based on the test dataset to assess the model's binary classification

effectiveness: the larger the AUC value, the better the model distinguishes between classes. When AUC >

0.8, the model prediction effect is very good (Sinha et al., 2008). In this study, both micro- and macro-

average ROC curves were plotted. The micro-average ROC curve aggregates the contributions of all

classes to compute the average ROC curve, treating each instance equally, while the macro-average ROC

curve computes the ROC curve for each class independently and then averages the results. These two

methods are commonly used for multi-class classification problems, but in this study, they were used to

give a more comprehensive comparison of model performance. The verification results are shown in

**Figure 6**. The AUC value of the CatBoost model reached 0.91, guaranteeing the performance and

prediction reliability of the CatBoost model. Based on this, the CatBoost model was selected to calculate

the flood risk in the YRDUA.

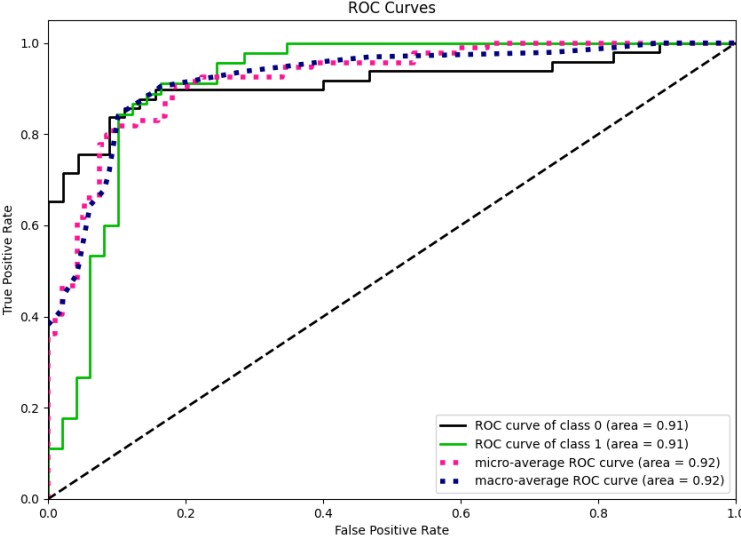

**Figure 6: Receiver operating characteristic (ROC) curves and corresponding area under the curve (AUC) values of the CatBoost model.**

493

**3.1.2 Importance and Interpretability Analysis of Hazard Factors**

In this study, the AutoML model was used specifically to assess flood hazard, which represents the

physical likelihood of flood occurrence and is directly driven by environmental factors such as rainfall,

topography, and drainage characteristics. Therefore, only the six second-level indicators under the hazard

dimension were used as input features in the AutoML model. This approach allowed us to focus the

model on identifying the key natural drivers of flooding, while the other dimensions — exposure,

vulnerability, and resilience—were later incorporated via the AHP method for comprehensive flood risk

evaluation.

To better understand the contribution of different hazard indicators to flood risk in the YRDUA, we

conducted both importance ranking analysis using the CatBoost model and interpretability analysis based

on SHAP.

The CatBoost model was used to quantify the relative importance of six key hazard indicators. The results,

shown in **Figure 7**, reveal significant differences in their influence. DEM was identified as the most

critical factor, contributing 68.55%, which far exceeds the other factors, which is also in line with the

findings of many researchers within the region (Mei et al., 2021; Wan et al., 2013). Low-lying areas

naturally function as water accumulation zones, increasing flood vulnerability. Additionally, urban areas

in the YRDUA are dominated by impervious surfaces, limiting infiltration and exacerbating flood risks.

While PREC is a primary factor in storm-induced flooding, its direct contribution to flood risk was

relatively low compared to DURA, which accounted for 10.07%. This highlights that the persistence of

extreme rainfall events is a stronger predictor of flood hazard than total annual precipitation.

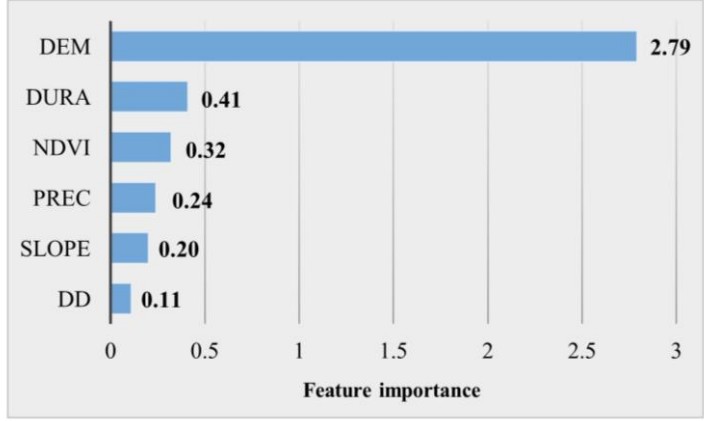


**Figure 7: Importance Ranking of Hazard Factors Based on the CatBoost Model.**



To further analyze the interpretability of the model and understand the impact of individual flood hazard
indicators on the model's classification results, this paper calculates Shapley Additive Explanations
(SHAP) to indicate the contribution of each feature in the sample (Lundberg and Lee, 2017). SHAP, a
game theory-based post-hoc interpretation method, quantifies the marginal impact of each feature on
model predictions. The SHAP summary plot in **Figure 8** (a) ranks features based on their absolute SHAP
values, consistent with the CatBoost importance ranking. Each row represents a feature, where red
indicates higher feature values and blue indicates lower values. The results show that DEM, SLOPE, and
NDVI negatively impact flood risk, meaning that higher elevation, steeper slopes, and greater vegetation
coverage reduce flood hazards. In contrast, DD, DURA, and PREC positively impact flood risk,
indicating that higher drainage density, longer durations of extreme rainfall, and increased precipitation
levels contribute to higher flood hazards. Among these, DEM has the highest absolute SHAP value, with
a strong clustering below zero, reinforcing its dominant role in flood risk determination.

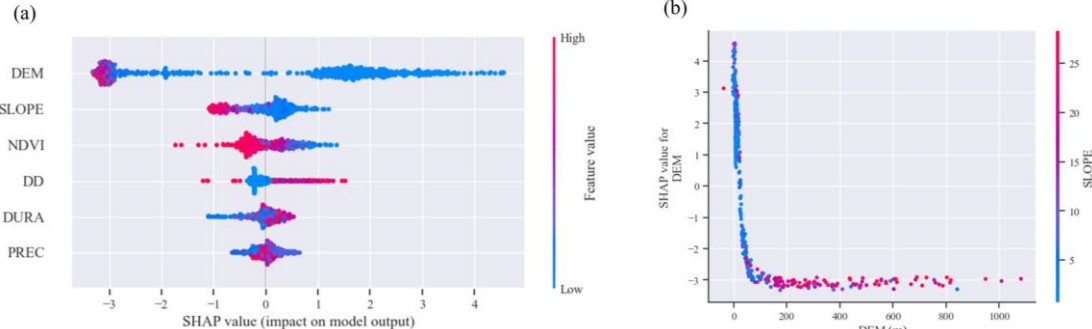

**Figure 8: (a) Scatter Plot of Hazard Indicators from SHAP Analysis. (b) SHAP Dual Dependence Analysis of Elevation and Slope Factors.**

To directly capture the interaction effects between paired indicator factors, this study used SHAP interaction values based on game theory, ensuring consistency while also explaining the interaction effects of individual predictions. For the DEM feature, which had the highest importance in the SHAP analysis, the factor most strongly correlated with it was SLOPE. Therefore, to illustrate how one feature interacts with another to affect the model training results, this study used DEM and SLOPE as examples to plot the SHAP interaction scatter plot, representing the dependency of the DEM feature. The results are shown in **Figure 8** (b). This dependency plot takes the form of a logarithmic function, indicating that as DEM increases, the flood hazard decreases. Additionally, the slope has a negative effect on the flood hazard in relation to elevation; that is, at lower elevations and gentler slopes, the flood hazard is greater.

**3.1.3 Determination of flood risk index weights**

A judgment matrix was constructed for 19 indicator factors. A hazard index was constructed based on feature importance calculated using AutoML. The exposure, vulnerability, and resilience indicators were determined based on existing literature and relevant expert scores (Hsiao et al., 2021). The judgment matrices were constructed using a hybrid approach. For the hazard indicators, feature importance scores generated from the AutoML model were used to inform the pairwise comparisons. For the exposure, vulnerability, and resilience indicators, the weights were determined by the authors based on a combination of expert judgments, a review of existing studies, and consideration of the local conditions

in the YRDUA. The Saaty 1–9 scale was applied to assign relative importance to each pair of indicators. Finally, the judgment matrix results were tested for consistency, and the CR value was 0.0058, i.e., << 0.100, indicating that the results passed the consistency test and that the flood risk index weight values calculated using the AHP were acceptable. The specific indicator weights and their corresponding impacts on flood risk are shown in Table 5. The "Attribute" column represents the impact of each indicator on flood risk, with "+" indicating a positive impact on flood risk and "-" indicating a negative impact on flood risk.

**Table 5: Flood risk index weights.**

| Dimension | Indicator | Unit | Attribute | Weight |
|---|---|---|---|---|
| Hazard (0.4798) | PREC | mm | + | 4% |
| | DURA | Day | + | 10.8% |
| | NDVI | | - | 7.6% |
| | DEM | km | - | 22.99% |
| | SLOPE | ° | - | 6.4% |
| | DD | km/km$^2$ | + | 3.2% |
| | AREA | km$^2$ | + | 1.1% |
| Exposure (0.1083) | DPOP | people/km² | + | 4.32% |
| | DGDP | 10,000 yuan/km² | + | 3.84% |
| | DBUI | km$^2$ | + | 1.16% |
| Vulnerability (0.1312) | PPOP_CHI | % | + | 4.92% |
| | PPOP_ELD | % | + | 3.04% |
| | PPOP_UEDU | % | + | 2.11% |
| | UR | % | - | 2.05% |
| Resilience (0.2807) | GDP per capita | 100 million yuan/10,000 People | - | 4.43% |
| | UEMP | % | + | 5.04% |
| | DOCS | Per person | - | 4.13% |
| | INSTS | Each | - | 0.45% |
| | BEDS | Per bed | - | 6.28% |

The weighted results reflect the degrees of influence of the different indicator factors on flood risk. Danger was the decisive factor affecting flood risk, with a weight of 0.4798, followed by resilience and vulnerability. Exposure had a relatively low impact on flood risk. In terms of danger, the topography and DURA were the main factors affecting the occurrence of flooding. These two indicators determined the characteristics of flood disasters in the YRDUA from the perspective of disaster-prone environments and driving factors, respectively. In terms of exposure, the YRDUA is a typical area with rapid social, economic, and population growths in China. High population and GDP densities increase the risk of flood exposure. In addition, the uneven age distribution and education levels of the population are important social factors affecting the risk of flood disasters in urban agglomerations. In terms of resilience, improving health and medical infrastructure, developing the regional economy, and reducing unemployment rates are conducive to improving the overall disaster response capacity of the region and reducing the risk of flood disasters in the YRDUA.

**3.1.4 Model results verification**

Based on CatBoost under the AutoML framework and AHP, the levels of flood hazard, exposure, vulnerability, and resilience were calculated for floods in the YRDUA and the spatial distribution of flood risks in the region was obtained according to the weights determined by the model. Combined with the natural breakpoint classification method, a flood risk zoning map of the YRDUA was constructed. The extracted flood points were superimposed on the map to verify whether the model exhibited good flood risk assessment capabilities. The results are shown in **Figure 9**, indicating that the distribution of flood points was consistent with the distribution of high and medium-to-high risk areas in the region, with the model assessment results corresponding well with the actual flooding situation. To specifically illustrate the correspondence of the results, the proportion of flood points distributed in high and medium-to-high risk areas was quantitatively calculated. The obtained value was 87.45%, indicating that the flood risk assessment results of the model in this study were highly credible, and subsequent analysis could be conducted.

As shown in **Figure 9**, the high and medium-to-high risk areas in the YRDUA were mainly located in the northern part of the region, concentrated in Chizhou, Anqing, Ma'anshan, and Xuancheng Cities in Anhui Province, Yancheng and Yangzhou Cities in Jiangsu Province, and Taizhou City in Zhejiang Province. Meanwhile, most areas of Hangzhou City had the lowest risk. The flood risks in cities such as Shanghai,

Nanjing, and Jinhua were also relatively low. The overall analysis showed that the flood risk in the study area was low in the southwest and high in the northeast, determined largely by natural terrain and meteorological factors. The spatial distribution of the flood hazard class was similar to the distribution of flood risks; exposure decreased stepwise from Shanghai to the surrounding areas, reflecting that densely populated and economically developed cities have higher exposure. Areas with higher vulnerability were mainly concentrated in Chizhou, Anqing, Xuancheng, Chuzhou, and Yancheng Cities. The number of vulnerable people in these cities was relatively high. Vulnerability has aggravated the flood risks in Chizhou and Anqing Cities on the basis of flood risk. Meanwhile, Shanghai had the best resilience performance, followed by those of Hangzhou, Suzhou, and Nanjing Cities, greatly lessening the flood risks in these cities.

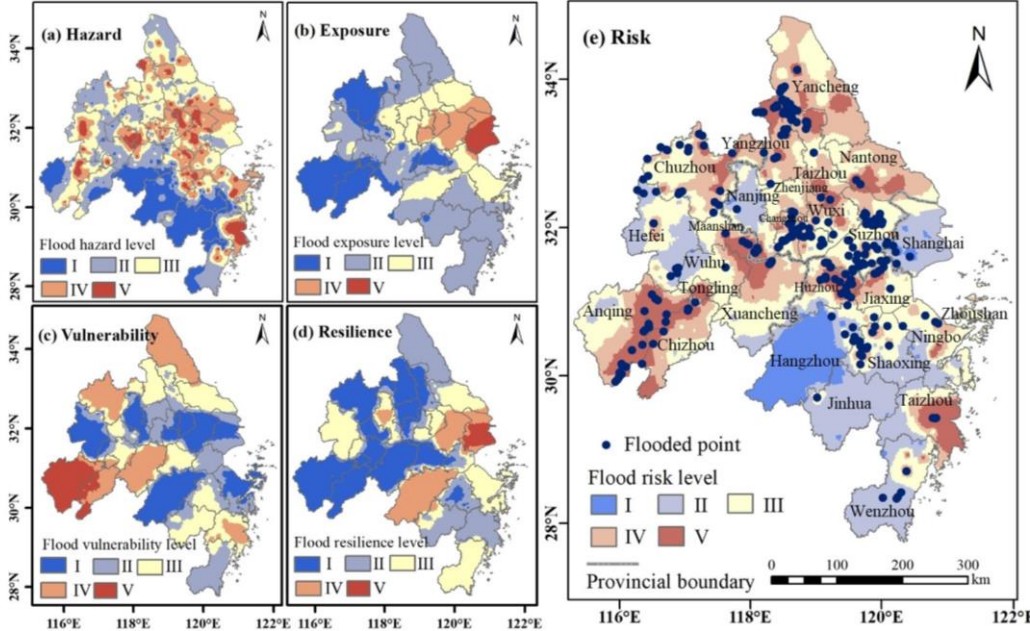

**Figure 9: Flood risk level distribution and verification results based on a flood risk assessment model. The flood hazard, exposure, vulnerability, and resilience of the YRDUA were calculated using CatBoost under the AutoML framework and AHP. The flood hazard level (a), flood exposure level (b), flood vulnerability level (c), flood resilience level (d), and flood risk spatial distribution (e) were derived through natural breaks classification in ArcGIS software based on model-determined weights, resulting in a flood risk zoning map for the Yangtze River Delta region.**

**3.2 Analysis of changes in the spatiotemporal characteristics of flood risk**

The flood risk results for the YRDUA from 1990 to 2020 were obtained based on the flood risk assessment model proposed in this study. The differences in flood risk among cities in the YRDUA over

the past few decades are primarily due to a complex interplay of various factors, including geographic
and climatic conditions, urbanization processes, socio-economic factors, ecological changes, and
historical flood events. The topography and precipitation patterns of different cities affect their capacity
for rainwater drainage and accumulation, while urbanization leads to an increase in impervious surfaces
and variations in infrastructure development, impacting flood management capabilities. Additionally,
differences in DPOP, economic development levels, and flood management policies can exacerbate flood
risk. Furthermore, the increasing frequency of extreme weather events due to climate change further
elevates flood risk. These factors determine the varying levels of flood risk among cities within the
YRDUA.
As the interannual difference in flood risk in the region was small and the change response was weak,
we selected the flood risk results for 1990, 2000, 2010, and 2020 to analyze the changes in the
spatiotemporal pattern. In this analysis, variables such as PREC and DURA exhibit clear temporal
variability, as they change year by year due to weather patterns. However, other factors like DEM,
SLOPE, NDVI, and urbanization indicators such as DPOP and GDP are spatial variables that do not
exhibit direct temporal changes, but their effects on flood risk are influenced by changing socio-economic
and ecological conditions.
Regarding spatial patterns (**Figure 10**), the flood risk in the YRDUA showed clear spatial heterogeneity.
The southwestern part of the study area and Shanghai have shown low flood risks over the past 30 years,
whereas the central and northern parts of the region have been more likely to face flood risks depending
on the natural conditions, population, economic conditions, and recovery capacity of the region.
Regarding temporal patterns, from 1990 to 2010, areas with high and medium-to-high risk decreased
markedly. By 2010, most of the YRDUA (except for a few areas) was in a state of medium risk or below,
with the southwestern region exhibiting a large range of low risk levels. The corresponding areas for
each risk level are shown in **Figure 11**. From 1990 to 2010, areas of low and low-to-medium risk levels
gradually increased, maximizing in 2010, whereas areas of medium risk and above continued to decrease.
By 2020, the number of high-risk areas for flooding increased. There is a tendency for areas of medium-
to-high risk in the central region to shift towards high-risk areas in 2020, as compared to the state in 1990.
Meanwhile, high-risk areas for floods also appeared in Chizhou and Anqing Cities in Anhui Province,
which was mainly due to the intensification of extreme weather, unbalanced population distribution, and
rapid economic development in recent years.

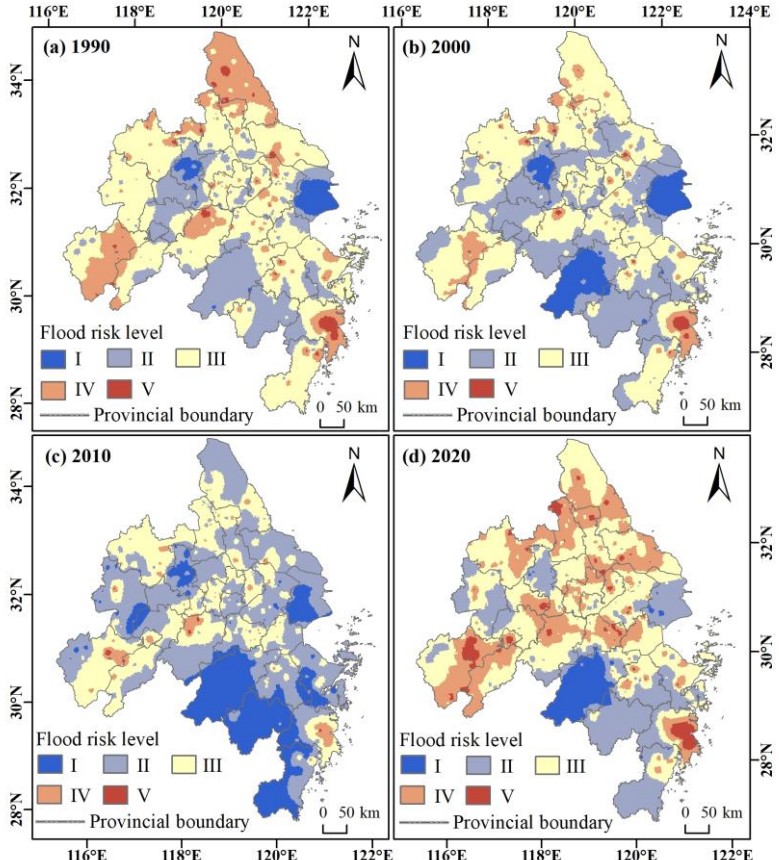


**Figure 10: Spatial distributions of flood risk in the YRDUA in different years during 1990–2020.**

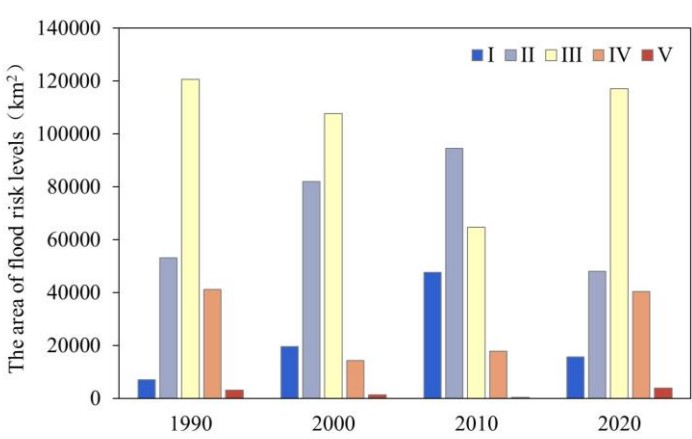


**Figure 11: Areas at different levels of flood risk in the YRDUA in different years during 1990–2020.**

To further analyze the changes in flood risk in the region, we calculated the change rate of the area of
different risk levels every 10 years and the overall change rate over 30 years. The interannual rate of
change was expressed in Eq. (13).
$R_{l,ij} = \frac{Risk_{l,j} - Risk_{l,i}}{Risk_{l,i}} \times 100\%,$ (13)

where $R_{l,ij}$ is the rate of change of the flood risk area of a certain level $l$ in a certain year, $i$ and $j$ are

different years, and $Risk_{l,j}$ and $Risk_{l,j}$ are the areas corresponding to the flood risk of this level in

different years.

The interannual variation rate of the flood risk is shown in **Table 6**. Results showed that the interannual

variation between the areas of low and high risk was relatively large. The low-risk area maximized in

2010, and both R $_{2000-1990}$ and R $_{2010-2000}$ showed a positive variation rate. The high-risk area showed the

largest interannual variation rate from 2010 to 2020, reaching 12.22% and causing the high-risk flood

area in 2020 to spread, resulting in a large high-risk area.

**Table 6: Interannual change rates of flood risk areas of different levels (expressed as percentages).**

|      | $R_{2000-1990}$ | $R_{2010-2000}$ | $R_{2020-2010}$ | $R_{2020-1990}$ |
|------|-----------------|-----------------|-----------------|-----------------|
| I    | 1.77%           | 1.44%           | -0.67%          | 1.21%           |
| II   | 0.54%           | 0.15%           | -0.49%          | -0.10%          |
| III  | -0.11%          | -0.40%          | 0.81%           | -0.03%          |
| IV   | -0.65%          | 0.25%           | 1.25%           | -0.02%          |
| V    | -0.53%          | -0.80%          | 12.22%          | 0.27%           |

Analyzing the flood risk of the entire urban agglomeration does not reveal the spatial scale effect of flood

risk, nor does it consider the correlation and impact of flood risk at different spatial scales. To reflect the

distribution of and changes in flood risk at different spatial scales within the region, the risk intensity of

different provinces was further analyzed, and the results are shown in **Figure 12**, respectively. In **Figure**

**12**, the average flood risk reflects the differences in risk development of the provincial administrative

units in Shanghai, Anhui, Zhejiang, and Jiangsu in terms of time and space. Overall, all administrative

units in the YRDUA exhibited the highest flood risk in 2020, and the overall risk trend increased. At the

provincial level, Shanghai's flood risk was consistently low, showing a trend of first decreasing from

0.152 in 1990 to 0.123 in 2000 and then gradually increasing to 0.311 in 2020. Among the other three

provinces, Jiangsu and Anhui had relatively high flood risks, reaching 0.525 and 0.516, respectively, in

2020, whereas Zhejiang had a relatively low flood risk, which remained stable between 1990 and 2010,

with no distinct changes.

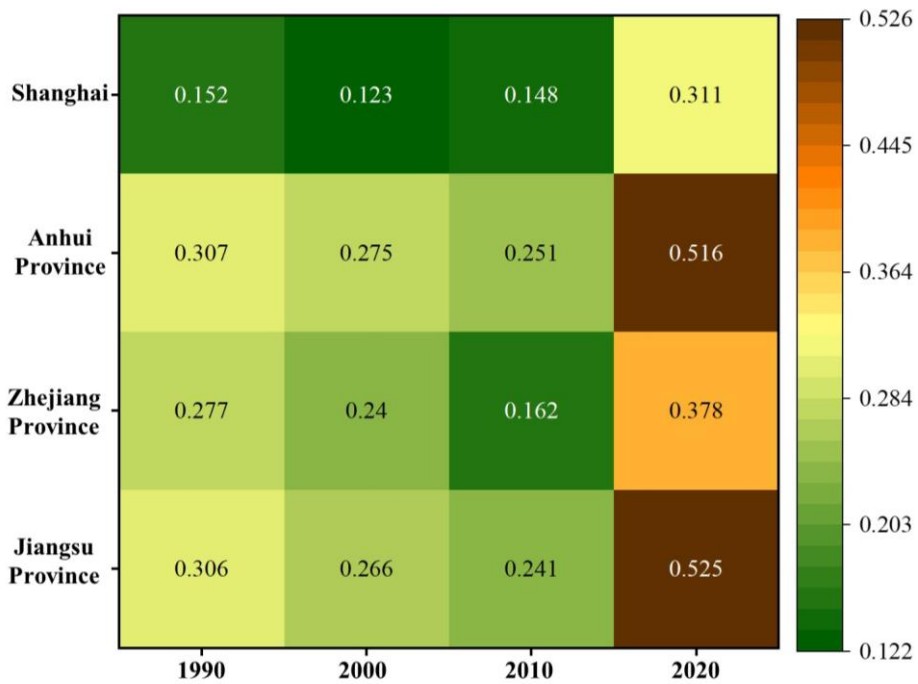


**Figure 12: Distribution of Average Flood Risk in Each Province of the Yangtze River Delta Urban**
**Agglomeration from 1990 to 2020.**

**4 Conclusion**
Flood risk assessment at the scale of urban agglomeration is a hot research topic in the field of disaster
prevention and mitigation. In this study, the flood risk assessment indexes for YRDUA were determined
in different dimensions of danger, exposure, vulnerability and resilience, and a flood risk assessment
model based on AutoML and AHP was constructed to study the changes of spatial and temporal
characteristics of flood risk in the region in the last 30 years from 1990 to 2020, aiming to provide
scientific basis for the prevention and resilience of the YRDUA. The main conclusions of this study are
as follows:
(1) In the flood risk calculation, the CatBoost model has the highest Precision, F1-score, and Kappa, and
its Precision can reach 0.9030. Further analysis of the ROC curve and the corresponding AUC value of
the model shows that its AUC value is 0.91, which indicates that the CatBoost model has the best
performance and prediction reliability. Therefore, the CatBoost model was selected to calculate the flood
risk in the YRDUA.

(2) Using the flood risk assessment model based on AutoML and AHP to obtain the flood risk of the YRDUA, superimposed on the flooded point data for comparative analysis, we found that the distribution of flooded points in the study area is basically consistent with the distribution of high and medium-to-high risk areas of flooding, and the proportion of the distribution of the quantification of its distribution is 87.45%, which indicates that the model in this study has a good performance and credibility regarding the assessment of flood risk.

(3) The spatial distribution of flood risk in the YRDUA during the 30-year study period shows obvious heterogeneity, with the southwestern part of the region and Shanghai City having a low flood risk, whereas the north-central part of the region faces a relatively high probability of flood risk. Between 1990 and 2010, there was a substantial decrease in the high and medium-to-high risk flood zones; yet by 2020, there was an increase in the high-risk flood zones. There is a tendency for the medium-to-high risk area in the center of the region to shift to a high-risk area, whereas high-risk areas also occur in the cities of Chizhou and Anqing in Anhui Province.

(4) All administrative units of the YRDUA exhibited the highest flood risk in 2020, with an overall trend of increasing risk. At the provincial level, Jiangsu and Anhui Provinces possess relatively high flood risks, whereas Zhejiang Province has a relatively low flood risk.

(5) The findings of this study provide valuable insights for flood risk management and policy-making. The flood risk maps generated in this study can serve as a scientific basis for urban planning, infrastructure development, emergency response, and disaster prevention strategies. By integrating these risk assessments into decision-making processes, government agencies and urban planners can optimize flood prevention measures and enhance regional resilience. Furthermore, the AutoML framework used in this study can be applied to other regions for flood risk assessment and can be integrated with future climate change scenarios to enable long-term forecasting and proactive disaster mitigation strategies.

**Data availability.**

Administrative boundaries and river network density were obtained from the Resource and Environmental Science and Data Center, Chinese Academy of Sciences (https://www.resdc.cn/DataList.aspx). Digital elevation data were derived from the SRTM1 dataset provided by USGS (https://earthexplorer.usgs.gov/). Land use data were sourced from the China Land

Cover Dataset (CLCD) developed by Wuhan University (https://zenodo.org/records/8176941). Hourly

precipitation data from 120 meteorological stations were obtained from the National Meteorological

Information Center, China Meteorological Administration (https://data.cma.cn/). Historical flood

inundation data were obtained from the MODIS-based Global Flood Database and validated using the

EM-DAT                                    disaster                                    database

(https://developers.google.com/earthengine/datasets/catalog/GLOBAL_FLOOD_DB_MODIS_EVENT

S_V1).

## Competing interests.

The authors declare that they have no competing financial interests or personal relationships that may

have influenced the work reported in this study.

## Author contributions.

**YG:** Writing - original draft preparation, Validation, Software, Methodology, Conceptualization **HL:** Writing-review & editing, Visualization, Supervision, Formal analysis. **YZ:** Methodology, Formal analysis. **HJ:** Writing - review & editing, Methodology. **SW:** Software, Formal analysis. **YG:** Visualization, Software. **SZ:** Writing-review & editing, Resources, Project administration, Funding acquisition, Conceptualization .

## Acknowledgements

This study was supported by the National Natural Science Foundation of China (Grant Nos. 42271483 and 42071364) and Jiangsu Provincial Natural Resources Science and Technology Project (JSZRKJ202405). We would like to thank Editage (www.editage.cn) for English language editing.

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
