# Peer review of "Evaluating Yangtze River Delta Urban Agglomeration"

_Natural Hazards and Earth System Sciences, 2024_

## Author Comment (AC2)

[Figure]

**Figure 2: The schematic map of the YRDUA.**

[Figure]

**Figure 3: (a) Flood inundation map of the study area. (b) Spatial Distribution of Flooded and Non-Flooded Points in the YRDUA.**

---

## Author Response (AR1)

**Dear Editor and Reviewers,**

**We sincerely appreciate the time and effort that the editor and both reviewers have dedicated to evaluating our manuscript. We are grateful for the insightful comments and constructive suggestions, which have significantly helped us improve the quality and clarity of our work. We have carefully considered each comment and have made the necessary revisions accordingly. Below, we provide detailed responses to each of the reviewers' suggestions and explain the corresponding modifications made in the manuscript.**

**Thank you again for your valuable feedback and support.**

**Reply on RC1**

1. Thank you for your valuable feedback. In response to your comment that the datasets were not sufficiently described, we have revised and improved Section 2.2.1 (Data Sources) to provide clearer and more detailed descriptions of the datasets used in this study.

Specifically, we have modified the table format and content to:

(1) Clarify dataset details (e.g., data processing, resolution, and coverage).

(2) Ensure consistency in formatting and descriptions across all dataset categories.

(3) Improve readability by structuring the information in a more accessible way.

The updated Table 1 now provides a more comprehensive overview of the datasets used, including spatial resolution, data preprocessing methods, and specific sources.

2. Thank you for your valuable feedback. In response to your comment regarding the level of detail in the results, we have carefully revised the manuscript and incorporated detailed modifications based on the specific comments you provided.

Each of your specific concerns has been addressed individually, with revisions made to enhance the clarity, completeness, and validity of the results. These modifications ensure that the findings are now presented with a greater level of detail, making them more transparent and easier to evaluate. We appreciate your insightful suggestions, which have significantly contributed to improving the manuscript.

3. The combination of machine learning and AHP methods for flood risk assessment is already quite common. However, using only a single machine learning algorithm tends to result in poor interpretability of flood risk, leading to uncertainty in the model's flood risk results. Additionally, further research is needed to efficiently and accurately select the optimal machine learning algorithm for the region.

Automatic machine learning tries to automatize the steps of feature extraction, model and algorithm selection, parameter optimization, and so on so that it needs no human assistance and avoids man-made bias. This approach only requires the configuration of different run times, allowing the algorithm to explore a wider array of model and parameter combinations within the allocated time, ultimately leading to the identification of the best-performing model.

This paper selects the Auto-Sklearn framework to address the binary classification problem of flooded point identification and to calculate flood risk. By utilizing the characteristics of automated machine learning, the efficiency of machine learning is improved, and the importance ranking of flood hazard factors is obtained. The next step is to use the AHP to calculate the relevant weights by combining the flood risk results with exposure, vulnerability, and resilience indicators. AHP aims to quantify the decision-making process by scoring weights according to their level of importance, ultimately yielding the flood risk results.

4. Line 57 - Replace the sentence in line 57 with: " Through continuous improvement and development of machine learning algorithms, ensemble methods have been addressed the limitations of traditional machine learning models. "

5. Line 76 - The meaning expressed in line 76 is inaccurate; a more precise phrasing would be: "The effectiveness of machine learning "automatically improves with experience," and a key challenge in the research is how to integrate the data processing capabilities and feature selection strengths of hybrid models with ensemble models."

6. Line 178 - The reason for choosing uneducated individuals as one of the indicators is that they are often part of socially vulnerable groups. People with lower levels of education may not fully understand warning information, disaster prevention measures, or have access to sufficient disaster

preparedness resources, which increases their vulnerability during disasters. Additionally, those with higher education levels typically have access to more information channels, while individuals with lower education levels may be unfamiliar with new technologies or information channels (such as mobile apps and internet alerts). At the same time, lower-educated groups may live in areas with less developed infrastructure, making it difficult for them to receive timely social aid and support. Uneducated individuals may also find it harder to regain economic independence after a disaster, as they may lack access to technical training or knowledge updates, leading to slower recovery. Although income level is an important factor in assessing vulnerability, average income distribution can sometimes obscure individual differences. For example, a region may have a high average income level, but low-income groups (such as uneducated individuals) can still be in a highly vulnerable state. Additionally, income level may not directly reflect an individual's awareness of disaster, knowledge reserves, or ability to take action.

7. Line 179- Urbanization rate refers to the proportion of the urban population to the total permanent population in a given region, and it reflects the level of urbanization in that area. This indicator has an inverse relationship with flood vulnerability. Generally speaking, the higher the urbanization rate of a region, the higher the level of social development and the capacity for protection, which can reduce flood vulnerability to some extent.

8. Line 187- The flood inventory map in this paper was developed using inundation data from the Global Flood Database and flood disaster data from the EM-DAT database, supplemented by satellite and Google image interpretation and verified against existing historical flood records. The actual flood-affected areas were delineated based on flood traces from the inundation datasets and image interpretations. For this study, 278 flood inundation points were randomly selected within the inundation data range during the study period, and the location of each point was used as the basis for subsequent statistical analysis of flood events.

9. Line 188- Thank you for your valuable feedback. To enhance clarity and better address your comment regarding Line 188, we have revised Section 2.2 in the manuscript.
The original Section 2.2 "Data Sources" has been renamed to "Section 2.2 Data Sources and

Processing" and is now divided into two subsections:

Section 2.2.1 Data Sources – Provides details on the datasets used in the study, including their sources and resolutions.

Section 2.2.2 Data Standardization and Preprocessing – Specifically addresses the issue raised in Line 188, explaining how features with different resolutions were mapped to meteorological stations to construct the training dataset.

In Section 2.2.2, we have added a detailed explanation of the (1) Unification of Spatial Scale and (2) Normalization of the Numerical Range:

(1) Unification of spatial scale means aligning data within the same coordinate range and resolution. The research data is standardized through projection transformation, converting all datasets into the same geographic and projected coordinate systems. The Kriging interpolation method is used to spatially process all discrete data. Finally, if the spatial data has different resolutions, resampling is performed to standardize all data to the same resolution, which in this study is unified to 30m×30m.

(2) Normalization of the numerical range can be achieved using a normalization process. Through a linear transformation, the values of the data are mapped to the range [0, 1], thus eliminating the influence of differing dimensions among the data indicators. In this study, the Min-Max Normalization method is used for normalization, and the formula is as follows:

$$x' = \frac{x - \min(x)}{\max(x) - \min(x)}$$

These modifications ensure that datasets from different sources and resolutions are properly standardized for flood risk assessment.

10. Line 194 - Data balancing was necessary in this study because an imbalanced flood and non-flood sample ratio led to biased model performance, where the classifier tended to favor the majority class. Through experiments, we found that using a 1:1 ratio for flooded and non-flooded points in the training dataset significantly improved the model's predictive performance, compared to 1:2 and 1:3 ratios, which resulted in decreased recall for the minority class. Therefore, we adopted a 1:1 sampling strategy to ensure a more balanced representation of flood and non-flood samples during training.

11. Line 201- Each label consists of 278 points representing entire dataset.

12. Line 201- The overview map of the study area has been revised to display the spatial distribution of flooded and non-flooded points.

13. Line 231- The text here contains a definition error, which has been corrected in the main body of the paper. Thank you to the reviewer for pointing this out.

14. Line 290- The definition of consistency has been added to the paper: In a pairwise comparison matrix, the decision-maker's judgments must exhibit logical coherence and transitivity. This means that if option A is considered more important than option B, and option B is considered more important than option C, consistency requires that option A must also be judged more important than option C.

15. The $\lambda$ in Eq. 11 should be $\lambda max$ as defined in Eq. 10, and this has been corrected in the paper.

16. Line 299 - This sentence has been revised: Where average Random Consistency Index (RI) represents the average random consistency which depends only on the order of the judgment matrix. The RI values for judgment matrices of order 1 to 10 are shown in Table 3.

17. Line 316 - Thank you for your valuable feedback. In response to your comment on Line 316, we have removed accuracy as an evaluation metric. Instead, we have adjusted the relevant sections throughout the manuscript to focus on Precision, Recall, F1-score, and the Kappa coefficient, which provide a more reliable evaluation of model performance in an imbalanced dataset.
We have carefully revised all occurrences of accuracy in the text to ensure consistency and clarity in our evaluation methodology. Please let us know if further modifications are needed.

18. Line 319 - The probability thresholds for accuracy, precision, recall, and F1-score range from [0, 1], while the Kappa coefficient ranges from [-1, 1].

19. Line 327 - We used 5-fold cross-validation to assess overfitting by comparing the performance of the training and testing sets. The experimental results indicate that the performance of the training set and testing set is relatively close, suggesting that the model does not exhibit overfitting.

20. Line 368 - Thank you for your valuable suggestion. We have adopted your recommendation and have cited Lundberg and Lee (2017) instead of Wang et al. (2023a) for the description of SHAP. Additionally, we have revised the sentence for improved clarity as follows:

"SHAP is an explanation method based on game theory and belongs to post-hoc model interpretation methods (Lundberg and Lee, 2017)."

21. Figure 8 - Thank you for your question regarding Figure 8. The figures were generated based on a flood risk assessment model that combines CatBoost under the AutoML framework and AHP to calculate flood hazard, exposure, vulnerability, and resilience in the YRDUA.

The flood hazard level (a), flood exposure level (b), flood vulnerability level (c), flood resilience level (d), and flood risk spatial distribution (e) were derived through natural breaks classification in ArcGIS software, using weights determined by the model. The final result is a flood risk zoning map for the YRDUA.

22. Line 450 - Thank you for your comment. We have revised the original text and incorporated the following content to clarify the causes of differences in flood risk over the past decades:

"The differences in flood risk among cities in the YRDUA over the past few decades are primarily due to a complex interplay of various factors, including geographic and climatic conditions, urbanization processes, socio-economic factors, ecological changes, and historical flood events. The topography and precipitation patterns of different cities affect their capacity for rainwater drainage and accumulation, while urbanization leads to an increase in impervious surfaces and variations in infrastructure development, impacting flood management capabilities. Additionally, differences in population density, economic development levels, and flood management policies can exacerbate flood risk. Furthermore, the increasing frequency of extreme weather events due to climate change further elevates flood risk. These factors determine the varying levels of flood risk among cities

within the YRDUA."

**Reply on RC2**

1. Line 13 - Thank you for your careful review. We appreciate your attention to detail. We have adopted your suggestion and revised "remains" to "remain" in Line 13 to ensure grammatical accuracy.

We appreciate your insightful comments and your help in improving the clarity and correctness of our manuscript.

2. Line 19 - Thank you for your suggestion. We have introduced CatBoost properly in the revised manuscript and have modified the sentence as follows:

"Results indicate that, among different assessment models, the Categorical Boosting (CatBoost) model achieves the highest accuracy, precision, F1-score, and kappa coefficient for flooded point identification."

This ensures that CatBoost is properly introduced before being referenced.

3. Thank you for your careful review. We have corrected all missing spaces before citations as mentioned and have thoroughly checked the entire manuscript to ensure consistency in citation formatting.

We appreciate your attention to detail and your valuable feedback in improving the clarity and presentation of our manuscript.

4. Line 78 - The point has been removed. Thank you for your correction.

5. Line 111-114 Thank you for your valuable suggestion. We have carefully revised Lines 108–114 to remove detailed results from the introduction and instead provide an overview of the manuscript structure. The revised section now reads:

"The comparative analysis of superimposed flooded points data shows a strong alignment between the distribution of flooded points in the study area and the high to medium-high risk areas,

highlighting the reliability and applicability of the proposed model. The remainder of this paper is structured as follows: Section 2 describes the study area, data sources, and methodology; Section 3 presents the results and analysis; Section 4 discusses the findings and their implications; and Section 5 concludes the study with key insights and recommendations."

6. Figure 2 - Thank you for your constructive feedback on Figure 2. I have updated the color palette to avoid any confusion with rivers and lakes, and have also inserted an inset map to illustrate the position of the YRDUA relative to the entire China. The Digital Elevation Model (DEM) units of measure, which are indeed meters above sea level (m asl), have been clearly stated. The revised figure is attached in the submission documents for your review. I appreciate your detailed suggestions and hope these revisions meet your expectations.

[Figure]

**Figure 1: The schematic map of the YRDUA.**

7. Table 1- Thank you for your valuable suggestions regarding Table 1. We have carefully considered your feedback and made the following improvements:

(1) Enhanced Dataset Descriptions – We have provided more detailed explanations for basic geographic information data to ensure clarity and completeness.

(2) Improved Table Readability – To enhance readability, we have inserted a horizontal dividing line after each data category, making it easier to distinguish between different dataset categories.

(3) Revised Table Caption – The table title has been updated to "Table 1: Description of the Datasets Used for Flood Risk Assessment, Their Characteristics, and Data Sources", better reflecting its comprehensive content beyond just a list of data sources.

These modifications ensure that dataset details are clearer, formatting is more structured, and the table is easier to interpret. We appreciate your detailed review and insightful recommendations, which have significantly improved the clarity and presentation of the dataset information.

**Table 1: Description of the Datasets Used for Flood Risk Assessment, Their Characteristics, and Data Sources.**

| Category | Details | Resolution | Data Source |
|---|---|---|---|
| Basic Geographic Information Data | Administrative boundaries and river network density data. | 30m | -Resources and Environmental Science and Data Center, CAS (https://www.resdc.cn/). -USGS (https://earthexplorer.usgs.gov/). -Wuhan University CLCD dataset (https://zenodo.org/records/8176941). - National Ecosystem Science Data Center (nesdc.org.cn). |
| | Digital Elevation Model (DEM) based on SRTM1 (30m), mosaicked and clipped to the study area (27 core cities). | | |
| | Land use data from CLCD (30m), includes 7 types: farmland, forest, shrubland, grassland, water, bare land, and impervious surfaces. | | |
| | NDVI data (2000–2020) calculated using the GEE platform. | | |
| Meteorological Data | Hourly precipitation data from 120 meteorological stations. Data preprocessed for outlier removal and missing value handling. | Station data | National Meteorological Information Center, China Meteorological Administration |
| Social Statistics | Population, unemployment, GDP, and healthcare statistics at the prefecture level. | Prefecture-level | Provincial and municipal statistical yearbooks and bulletins |

| | Urbanization rate calculated using urban population proportion. | | |
| :--- | :--- | :---: | :--- |
| | GDP density and per capita GDP derived from total GDP and land area/population. | | |
| Historical Disaster Data | Flood inundation data from the MODIS-based Global Flood Database (2000–2018), processed to focus on the YRDUA region. To ensure comprehensive selection of inundation points, the inundated areas within the time frame were overlaid to produce a historical flood map. | 250m | Global Flood Database (https://www.emdat.be/). |

8. Lines 168-183 Thank you for your insightful comment regarding the definition of heavy rainfall in Line 172. Considering its importance as a key indicator in our analysis, we have revised Lines 168-183 to provide a clearer and more precise definition.

The revised section now explicitly states:

(1) A heavy rainstorm event is defined according to the Meteorological Bureau's criteria as rainfall of 50mm or more within 24 hours.

(2) Annual Cumulative Heavy Rainfall Duration (DURA) is defined as the total number of days in a year when heavy rainstorm events occur at meteorological stations within the study area.

(3) The explanation now clarifies the relationship between prolonged heavy rainfall duration and flood risk, reinforcing why DURA is a more crucial factor than total annual precipitation in flood hazard assessment.

The revisions are as follows:

The hazard indicators consisted of six indices: Average annual precipitation (PREC), Annual Cumulative Heavy Rainfall Duration (DURA), Digital Elevation Model (DEM), SLOPE, Drainage Density (DD), and Normalized Difference Vegetation Index (NDVI). Rainfall is the primary factor leading to flooding, particularly extreme rainstorms caused by climate change. According to the Meteorological Bureau's definition, a heavy rainstorm event is characterized by rainfall of 50mm or more within 24 hours. DURA is defined as the total number of days with heavy rainstorm events occurring at all meteorological stations within the study area each year. The more days heavy rainstorms accumulate and the longer their duration, the greater the likelihood of flooding and other

disaster events. DEM and SLOPE are important topographical indicators. Areas with low DEM and SLOPE values are generally more susceptible to flood threats. DD refers to the area of rivers or lakes per unit of land surface area and is a crucial indicator of a watershed's structural characteristics. It determines the watershed's capacity to withstand flooding. The higher the DD, the greater the likelihood of flooding and the higher the potential flood risk. Vegetation plays a role in water storage, retention, and infiltration. The lower the vegetation coverage, the weaker the buffering capacity, making it more likely for surface water to accumulate. The NDVI index measures the relative abundance of green vegetation, with values ranging from -1 to 1. The higher the value, the greater the vegetation coverage, and the lower the risk of flooding.

Land area (AREA), Population Density (DPOP), GDP Density (DGDP), and Building Density (DBUI) were selected as exposure indicators to assess the degree of vulnerability of both the natural environment and social systems to flooding. The land area for each administrative unit at the prefecture-level city is calculated individually. A larger land area corresponds to a greater extent exposed to flooding. DPOP and DGDP represent the concentration of population and assets, respectively. Areas with higher DPOP and DGDP are more susceptible to potential threats from pluvial flooding. DBUI, the ratio of total building area to total land area in a region, indicates the building density. A higher DBUI reflects greater exposure to flooding.

Vulnerability indicators focus more on the social aspects of flood disasters. This study selects four vulnerability indicators: Proportion of Child Population (PPOP_CHI), Proportion of Elderly Population (PPOP_ELD), Proportion of Uneducated Population (PPOP_UEDU), and Urbanization Rate (UR). Age is a key feature of social vulnerability, and both the population aged 0-14 and those over 65 are considered vulnerable groups, as these age groups are more susceptible to flood damage. The uneducated population generally has a weaker awareness of disaster risks and lower self-protection capacity, which makes this group more vulnerable to flooding. The urbanization rate refers to the proportion of the urban population in the total resident population of a region. This indicator is inversely related to flood vulnerability. In general, a higher urbanization rate indicates greater social development and stronger protective capacities, which can reduce vulnerability to flooding to some extent.

The resilience indicators selected in this study include Gross Domestic Product (GDP) per capita, Unemployment Rate (UEMP), Number of Doctors (DOCS), Number of Medical Institutions

(INSTS), and Number of Hospital Beds (BEDS). GDP per capita is the ratio of a region's GDP to its total resident population over a specified period, reflecting the region's economic condition. A higher GDP per capita indicates a more developed economy, which is associated with a greater capacity to recover quickly after a flooding event. The Unemployment Rate (UEMP) measures the proportion of the idle labor force, indirectly reflecting the stability of urban development. A high unemployment rate signals economic instability, which weakens the capacity to cope with floods and extends the time required for post-disaster recovery, thus impeding disaster response efforts. The indicators of DOCS, INSTS, and BEDS provide insights into a region's healthcare and medical support capabilities. Areas with stronger healthcare systems are better positioned to manage flood risks and recover more effectively from such disasters.

9. Thank you for your insightful question regarding the inclusion of three indicators related to the sanitary sector. These indicators—doctors, medical institutions, and hospital beds—were selected to comprehensively capture the region's healthcare capacity, which plays a crucial role in resilience during and after disasters.

While all three indicators pertain to healthcare, each represents a distinct aspect of flood resilience:
(1) Doctors reflect the availability of medical personnel to provide immediate care.
(2) Medical institutions indicate the infrastructure of healthcare facilities, which is essential for disaster response.
(3) Hospital beds measure the capacity to accommodate affected individuals, particularly in large-scale flood events.

Together, these indicators provide a balanced and multidimensional assessment of how the healthcare sector contributes to flood resilience.

Regarding the suggested inclusion of civil protection forces, law enforcement, and firefighters, we acknowledge their importance in disaster response. However, data availability constraints prevent us from including these indicators in our analysis. The relevant data for civil protection and emergency response forces are only available after 2012, whereas our study covers the years 1990, 2000, 2010, and 2020. Due to this limitation, we prioritized indicators with consistent data availability across all study periods.

Furthermore, previous research has demonstrated the importance of healthcare-related indicators in flood risk assessments. Ekmekcioğlu et al., (2021) proposed a hierarchical procedure that incorporates multiple flood vulnerability and hazard criteria, including healthcare capacity, to generate district-based flood risk maps. The study highlights that integrating healthcare infrastructure in flood risk assessments improves the ability to quantify social vulnerability and disaster mitigation capacity.

Additionally, research by Ahmed et al., (2022) in Geocarto International emphasized that mitigation capacity is a critical component in spatial flood risk mapping. The study found that regions with better healthcare infrastructure exhibit enhanced resilience and faster recovery from flood disasters. The inclusion of doctors, medical institutions, and hospital beds aligns with these findings, as they directly contribute to flood preparedness, emergency response efficiency, and post-disaster recovery. Based on these studies, we believe that healthcare-related indicators are essential for evaluating community resilience in flood risk assessments. While we recognize the role of emergency response units, our choice of indicators ensures consistency across different time periods and provides a comprehensive understanding of flood resilience.

10. Thank you for your insightful question regarding the division of the training and testing datasets. Based on your feedback, we have revised the manuscript to provide a clearer explanation of this process.

(1)Clarified Historical Disaster Data in Section 2.2

In Section 2.2, we have provided a more detailed explanation of the historical disaster data, specifically the flood inundation data from the MODIS-based Global Flood Database (2000–2018). This dataset was processed and cropped to focus on the Yangtze River Delta Urban Agglomeration (YRDUA).

(2)New Subsection 2.3: "Extraction of Historical Flood Inundation Points"

To explicitly address the division of the training and validation datasets, we have added a new subsection (2.3) titled "Extraction of Historical Flood Inundation Points", positioned before the original "2.3 Establishment of a Flood Risk Assessment Indicator System".This section explains how the historical flood map for the study area was generated and the criteria used to extract and separate the flooded and non-flooded points.To further illustrate this process, we have included Figure 3:

[Figure]

Figure 3: (a) Flood inundation map of the study area. (b) Spatial Distribution of Flooded and Non-Flooded Points in the YRDUA.

(3)Revised Section Numbering for Better Flow

The original Section 2.3 has been renumbered as Section 2.4 ("Establishment of a Flood Risk Assessment Indicator System"), and the order of the subsequent sections has been adjusted accordingly to maintain logical coherence in the manuscript.

11. Thank you for your insightful comments regarding the model names used in the manuscript. Based on your feedback, we have revised the model names to ensure clarity and accuracy.

(1) "Linear" has been updated to "Linear Regression" to explicitly specify that this refers to the linear regression model.

(2) "Neural Network" has been updated to "Multi-layer Perceptron (MLP) Neural Network" to clarify the specific type of neural network used in the study.

12. Section 2.4: Thank you for your insightful comments regarding data processing and spatial resolution. To enhance clarity and better address your concerns, we have made the following revisions in Section 2.2 of the manuscript.

(1) Renaming and Restructuring Section 2.2

(a) The original "Section 2.2 Data Sources" has been renamed "Section 2.2 Data Sources and Processing" to better reflect both the datasets and the processing steps.

(b)This section is now divided into two subsections:

Section 2.2.1 Data Sources – Provides details on the datasets used in the study, including their sources and resolutions.

Section 2.2.2 Data Standardization and Preprocessing – Addresses how features with different resolutions were mapped and standardized for analysis.

(2) Clarification on Raster Data and Resolution Standardization

(a) The analysis was performed using raster data, and we ensured that all datasets were standardized to a uniform resolution of 30m × 30m before model training.

(b) The preprocessing workflow involved projection transformation, spatial interpolation, and resampling to align all datasets within the same coordinate reference system and spatial scale.

(3) Detailed Explanation in Section 2.2.2

In Section 2.2.2, we have explicitly explained the two key preprocessing steps:

(a) Unification of Spatial Scale:

Data were standardized through projection transformation to ensure that all datasets were aligned within the same geographic and projected coordinate system. To process discrete spatial data, the Kriging interpolation method was applied, ensuring a smooth and continuous representation. Additionally, for datasets with different resolutions, resampling was performed to standardize them to a uniform 30m × 30m resolution for consistency in analysis

(b) Normalization of the Numerical range

A Min-Max Normalization process was applied to scale all values to [0,1], ensuring that different feature dimensions do not introduce bias into the model. The formula is as follows:

$$x' = \frac{x - \min(x)}{\max(x) - \min(x)}$$

13. Line 231-232 The text here contains a definition error, which has been corrected in the main body of the paper. Thank you to the reviewer for pointing this out.

14. Thank you for your suggestion regarding Equation 5. Based on your feedback, we have modified the notation to subscript "score" as recommended. We appreciate your careful review and constructive input, which have helped improve the clarity and consistency of the manuscript.

15. Thank you for your attention to detail in reviewing the manuscript. I have updated "judgements" to "judgments" in Table 2 as you suggested. I appreciate your guidance on this matter.

16.Line 299: This sentence has been revised: Where average Random Consistency Index (RI) represents the average random consistency which depends only on the order of the judgment matrix. The RI values for judgment matrices of order 1 to 10 are shown in Table 3.

17. Section 3.1.1: Thank you for your comment. We have added the training phase results in Section 3.1.1 and updated **Table 4** to include both training and test set performances. The results show that most models performed well on the training set but experienced a decline on the test set, indicating variations in generalization ability. CatBoost demonstrated strong robustness, while models like Decision Tree and Nearest Neighbors showed a more significant drop, suggesting higher sensitivity to overfitting.

18. Thank you for your comment. We have made the requested changes and added an explanation of the micro- and macro-average ROC curves in the text to clarify their meaning and relevance in our analysis.

19. Thank you for your valuable feedback. Based on your suggestion, we have made revisions to Section 3.1.2 to improve clarity and coherence. The two subsections—"Ranking of Importance" and "SHAP Interpretability Analysis"—have now been integrated into a single, more cohesive section titled "Importance and Interpretability Analysis of Hazard Factors".
In this revised section, we first present the importance ranking of the key flood hazard indicators using the CatBoost model and then follow with an in-depth explanation of the SHAP analysis. We also clarified the use of SHAP interaction values to capture the interaction effects between key features, specifically DEM and SLOPE, which was highlighted through the SHAP dependency plot. These revisions aim to ensure that the content flows more logically and provides a more integrated discussion of the analysis. We hope this addresses your concern and improves the overall clarity of the manuscript.

20. Thank you for your suggestion regarding the sentence in lines 354-355. I have rewritten the sentence for clarity and conciseness in the manuscript.

21. Figure 7 (b): Thank you for your helpful comment. The unit is now included in the figure.

[Figure]

**Figure 2: (a) Scatter Plot of Hazard Indicators from SHAP Analysis. (b) SHAP Dual Dependence Analysis of Elevation and Slope Factors**

22. L404 and Table 5: Thank you for your valuable feedback. To clarify the meaning of the "Attribute," we have made the following revisions:

The specific indicator weights and their corresponding impacts on flood risk are shown in Table 5. The "Attribute" column represents the impact of each indicator on flood risk, with "+" indicating a positive impact on flood risk and "-" indicating a negative impact on flood risk.

23. Thank you for your valuable feedback. In response to your suggestion, we have added a new section 2.7 "Determination of Flood Risk Levels" in the methodology, where we provide a detailed explanation of the natural breakpoint classification method.

24. Thank you for your inquiry regarding the resolution of the rasters used in Figures 8 and 9. The raster data for these figures were obtained at a resolution of 30m x 30m.

Additionally, to clarify the data preprocessing steps, we have added an explanation in Section 2.2.2, "Data Standardization and Preprocessing."

25. The following content has been added to Section 3.2: In this analysis, variables such as PREC

and DURA exhibit clear temporal variability, as they change year by year due to weather patterns. However, other factors like DEM, SLOPE, NDVI, and urbanization indicators such as DPOP and GDP are spatial variables that do not exhibit direct temporal changes, but their effects on flood risk are influenced by changing socio-economic and ecological conditions.

26. Line 483 - Thank you for your suggestion. We have revised the percentage change values to retain two decimal to improve readability and clarity.

27. Thank you for your valuable suggestion. We have revised **Table 6** to explicitly indicate that the change rates are expressed as percentages. Additionally, we have updated the table caption and added percentage signs (%) to all values for clarity.

28. Thank you for your valuable suggestion. To address your comment, we have added point (5) in the conclusion section, highlighting the practical implications of our findings for flood risk management and policy-making. This addition discusses how the flood risk maps generated in this study can serve as a scientific basis for urban planning, infrastructure development, emergency response, and disaster prevention. It also emphasizes the potential of integrating these assessments into decision-making processes to enhance flood prevention measures and regional resilience. Furthermore, the AutoML framework used in this study can be applied to other regions and incorporated with future climate change scenarios for long-term forecasting and proactive disaster mitigation planning.

**Reference**

Ahmed, N., Hoque, M. A.-A., Howlader, N., and Pradhan, B.: Flood risk assessment: role of mitigation capacity in spatial flood risk mapping, Geocarto International, 37, 8394–8416, https://doi.org/10.1080/10106049.2021.2002422, 2022.

Anon: A weighted metric scalarization approach for multiobjective BOHB hyperparameter optimization in LSTM model for sentiment analysis, Information Sciences, 644, 119282, https://doi.org/10.1016/j.ins.2023.119282, 2023a.

Anon: Efficient LBP-GLCM texture analysis for asphalt pavement raveling detection using eXtreme Gradient Boost, Construction and Building Materials, 401, 132731,

https://doi.org/10.1016/j.conbuildmat.2023.132731, 2023b.

Ekmekcioğlu, Ö., Koc, K., and Özger, M.: District based flood risk assessment in Istanbul using fuzzy analytical hierarchy process, Stoch Environ Res Risk Assess, 35, 617–637, https://doi.org/10.1007/s00477-020-01924-8, 2021.

Omar, A. and Abd El-Hafeez, T.: Quantum computing and machine learning for Arabic language sentiment classification in social media, Sci Rep, 13, 17305, https://doi.org/10.1038/s41598-023-44113-7, 2023.

---

## Author Response (AR2)

**Dear Editor and Reviewers,**

**We sincerely appreciate the time and effort that the editor and reviewers have devoted to reviewing our manuscript. We are truly grateful for your insightful comments and constructive suggestions, which have greatly contributed to improving the overall quality, clarity, and rigor of our work. We have carefully addressed each point raised and revised the manuscript accordingly. Below, we provide detailed, point-by-point responses to all reviewer comments, along with explanations of the modifications made to the manuscript.**

**Thank you once again for your valuable feedback and continued support.**

**Editor Comment**

a) Figure 2 may contain a territory that is disputed according to the United Nations (small map in the upper left corner).

Response: Thank you for pointing this out. We have replaced the map to address this issue. The upper-left inset map in Figure 2 has been changed: it now shows the location of the Yangtze River Delta Urban Agglomeration (YRDUA) within the Euro-Asia continent, without including any disputed territories. The position of the YRDUA is indicated using a red five-pointed star symbol, ensuring both clarity and compliance with the journal's map policies.

[Figure]

**Figure 1: The schematic map of the YRDUA.**

b) Section "Author contribution": please use initials for the authors' names.

Response: Thank you for your suggestion. We have revised the Author Contributions section to use initials for all authors' names, in accordance with the journal's formatting requirements.

The revised author contributions are as follows:

**YG:** Writing - original draft preparation, Validation, Software, Methodology, Conceptualization **HL:** Writing-review & editing, Visualization, Supervision, Formal analysis. **YZ:** Methodology, Formal analysis. **HJ:** Writing - review & editing, Methodology. **SW:** Software, Formal analysis. **YG:** Visualization, Software. **SZ:** Writing-review & editing, Resources, Project administration, Funding acquisition, Conceptualization .

**Reply on RC1**

1. Figure 2. Please change m to m asl, as I suggested before.

Response: Thank you very much for your constructive suggestions.

We have made the following modifications to Figures 2:

(1) The elevation unit has been corrected from "m" to "m asl" (meters above sea level) **in Figure 2**.
(2) The inset map in the upper-left corner has been replaced: it now shows the position of the YRDUA within the Euro-Asia continent, and no longer includes any disputed territories.
(3) The location of the study area is now marked with a red five-pointed star, making it easily identifiable.

[Figure]

**Figure 2: The schematic map of the YRDUA.**

2. L165: please add more information about which kind of kriging you use and information about the semivariogram adopted.

Response: Thank you for your helpful suggestion. We have revised the manuscript to specify the type of Kriging interpolation used and the semivariogram model employed.

In this study, we applied ordinary Kriging, which assumes a constant but unknown mean within the local neighborhood. The semivariogram model adopted was the spherical model, which is widely used in environmental geostatistics due to its smooth and bounded nature.

This clarification has been added to the revised manuscript in Section 2.2 (Data Preprocessing) as follows:

"To generate continuous spatial surfaces from discrete data points, we applied the Ordinary Kriging interpolation method, which assumes a constant but unknown local mean (Cressie, 1990). A spherical semivariogram model was adopted to capture spatial autocorrelation, as it is widely used in environmental geostatistics for its bounded range and smooth continuity (Webster and Oliver, 2007). The interpolation process was carried out using ArcGIS 10.8."

We believe this addition improves methodological transparency and will help readers better understand the spatial processing procedure.

3. I'm curious about why the testing performance in several metrics is better in the testing dataset (e.g., extra trees or decision trees) than with the training set. The authors should try to explain that strange behavior.

Response: Thank you for this insightful observation. We agree that it is uncommon for a model to perform better on the test set than on the training set. However, we believe this phenomenon in our study is justifiable due to several technical and methodological considerations, as detailed below.

In our study, AutoML (Auto-sklearn) was employed to construct a binary classification model for flood hazard prediction. Only the six second-level hazard indicators were used as input features: Average Annual Precipitation (PREC), Annual Cumulative Heavy Rainfall Duration (DURA), Digital Elevation Model (DEM), Slope (SLOPE), Drainage Density (DD), and Normalized Difference Vegetation Index (NDVI). These variables represent the natural drivers of flood events. The classification task was based on 556 points (278 flooded + 278 non-flooded) selected from the historical inundation areas, verified using remote sensing and flood databases. The sample set was split into a 70% training set and a 30% test set (as detailed in Section 2.3 and Section 3.1.1).

Regarding your question, several technical factors may explain the slightly better performance observed in some metrics on the test set:

(1) Small Sample Size and Statistical Variability:

The total sample size (556) is relatively limited. With a test set of ~167 samples, minor variations (e.g., one or two easier-to-classify cases) can noticeably influence performance metrics like precision or F1-score. These fluctuations are within the range of statistical randomness and are not indicative of data leakage or overfitting.

(2) Random Partitioning and Class Balance:

Although the dataset was balanced (1:1 flooded to non-flooded), the training and test sets were split randomly. This could lead to the test set containing slightly "easier" or more representative instances, whereas the training set might include a few borderline or noisy samples. Thus, the model may generalize better to the test set purely by chance.

(3) Regularization and AutoML Behavior:

All models were trained under AutoML's hyperparameter tuning pipeline, which uses internal cross-validation and enforces regularization (e.g., tree depth limits or minimum leaf size). As a result, the model may slightly underfit the training data to avoid overfitting. This behavior, particularly with models like Extra Trees that average multiple randomized trees, can result in slightly higher test performance under certain conditions.

(4) Evaluation Metric Sensitivity:

Precision, recall, and F1-score are sensitive to class distribution and small sample shifts. A few additional true positives or fewer false positives in the test set can raise these metrics, especially in a dataset of this size.

(5) No Data Leakage:

We confirm that the training/test split was performed before any modeling or hyperparameter tuning, and all parameter optimization was confined to the training set. The test set was never seen during training or model selection, ruling out data leakage. If leakage had occurred, we would expect all metrics to be uniformly higher—not just marginally so in a few models.

(6) Consistency with Other Studies:

Similar patterns of test performance exceeding training performance in individual metrics have been observed in other flood-related studies. For instance, Wang et al. (2024) reported a comparable result in their study on the Spatio-temporal evolution of public opinion on urban flooding during the 7.20 Henan extreme flood event.

To clarify this in the manuscript, we have added the following sentence to Section 3.1.1 (AutoML optimal model selection):

"Interestingly, in a few cases (e.g., Extra Trees), test set performance slightly exceeded that of the training set in certain metrics. This is not uncommon in small, balanced datasets and may result from a combination of factors such as random sampling variation, slightly easier test samples, or appropriate regularization that reduces overfitting in the training set."

In conclusion, the slight outperformance on the test set is a result of normal statistical variability, model regularization, and careful design of a balanced dataset. This behavior is well-documented in prior literature and does not indicate methodological issues.

4. Data availability statement. Please take care of this section and write all the availability statements, as I requested before. Nowadays, it is not enough to write "Data will be made available on request."

Response: The revised "Data availability" section now explicitly lists each dataset and its corresponding provider, along with web links to public repositories or official data platforms where applicable. We have also removed the previous general statement "Data will be made available on request," as advised.

Administrative boundaries and river network density were obtained from the Resource and Environmental Science and Data Center, Chinese Academy of Sciences (https://www.resdc.cn/DataList.aspx). Digital elevation data were derived from the SRTM1 dataset provided by USGS (https://earthexplorer.usgs.gov/). Land use data were sourced from the China Land Cover Dataset (CLCD) developed by Wuhan University (https://zenodo.org/records/8176941). Hourly precipitation data from 120 meteorological stations were obtained from the National Meteorological Information Center, China Meteorological Administration (https://data.cma.cn/). Historical flood inundation data were obtained from the MODIS-based Global Flood Database and validated using the EM-DAT disaster database (https://developers.google.com/earthengine/datasets/catalog/GLOBAL_FLOOD_DB_MODIS _EVENTS_V1).

**Reply on RC2**

1. Line 57 - What are the limitations that have been addressed by ensemble methods? What are ensemble methods?

Response: Thank you for this helpful comment. In the revised manuscript, we have clarified the definition of ensemble methods and explicitly listed the limitations of traditional machine learning models that ensemble methods aim to address. Specifically, we now explain that ensemble methods are a class of ML techniques that combine multiple base learners to form a stronger predictive model, and they are designed to mitigate issues such as high variance,

overfitting, sensitivity to noise, and poor generalization.

The revised sentence now reads:

"Ensemble methods are a class of machine learning techniques that combine multiple base learners to form a stronger predictive model (Webb and Zheng, 2004). They are designed to overcome several limitations of individual models, such as high variance, overfitting, sensitivity to noise, and poor generalization(Yang et al., 2013). By aggregating the outputs of weak learners, ensemble methods significantly enhance model stability, accuracy, and robustness—especially in high-dimensional and complex classification or regression tasks (Kazienko et al., 2015)."

2. Line 58 - What are integrated ML methods?

Response: Thank you for pointing out this ambiguity. In the original manuscript, the term "integrated ML methods" was intended to refer to ensemble machine learning techniques. To avoid confusion, we have revised the sentence to use the more accurate and standard term "ensemble ML techniques." Additionally, we have provided specific examples of ensemble methods such as bagging (Random Forest), boosting (XGBoost, CatBoost), and stacking, and clarified their relevance to hydrological modeling.

The revised sentence reads:

"Various ensemble ML techniques, including bagging (e.g., Random Forest), boosting (e.g., XGBoost, CatBoost), and stacking, have been widely used in hydrology, with boosting algorithms in particular showing strong performance in flood prediction and risk assessment (Shafizadeh-Moghadam et al., 2018; Mirzaei et al., 2021; Yan et al., 2024)."

3. Line 71 - The statement is unclear and seems incorrect - the authors have themselves pointed out that by carefully configuring ML models, they can in fact perform well across varied problems.

Response: Thank you for your careful reading and insightful comment. We acknowledge that the original statement may appear contradictory. In the revised version, we have clarified our point by emphasizing that while machine learning algorithms can achieve strong performance in many cases, no single algorithm universally outperforms others across all tasks. Therefore, careful configuration of each component in the ML pipeline—such as feature engineering, model selection, and hyperparameter tuning—is essential to adapt to different problems.

The revised sentence reads:

"While machine learning algorithms have demonstrated strong performance in many domains, no single algorithm consistently performs best across all types of problems. Therefore, to achieve optimal performance, it is essential to carefully configure key components of the ML

pipeline, including feature engineering, model selection, and hyperparameter tuning (Li et al., 2017; Raschka, 2020)."

4. Line 76 - What does experience mean?

Response: Thank you for your insightful comment. We have revised the sentence to explicitly define what "experience" refers to in the context of machine learning. We also revised the structure of the sentence for clarity and to maintain a logical flow into the key research challenge addressed in our study.

The updated sentence now reads:

"The effectiveness of ML improves with experience, where "experience" refers to the model's iterative exposure to training data and its ability to learn patterns from labeled examples (Jordan and Mitchell, 2015; Nagarajah and Poravi, 2019). One key challenge addressed in this study is how to automatically optimize model components such as feature selection and algorithm configuration in flood risk prediction, while maintaining high accuracy and adaptability across complex hydrological conditions. "

5. Line 80 - What are some of the various problems being addressed by AutoML?

Response: Thank you for your helpful comment. We agree that the original wording was too general and did not adequately specify the kinds of problems AutoML addresses. In the revised manuscript, we now explicitly state that AutoML automates several key processes in the machine learning pipeline, including feature selection, model selection, hyperparameter tuning, and ensemble learning. These steps are critical to improving model performance and reducing subjectivity.

In addition, we have clarified the relevance of AutoML in the context of this study. Specifically, AutoML enables automatic optimization of hazard factor selection, model construction, and parameter adjustment in the flood risk assessment workflow.

The revised passage reads:

"AutoML is an innovative machine learning framework that automates key stages of the model development pipeline, including feature selection, model selection, hyperparameter tuning, and ensemble learning. By addressing these challenges, AutoML reduces reliance on expert knowledge and minimizes subjectivity in model building (He et al., 2021; Consuegra-Ayala et al., 2022). In the context of this study, AutoML enables the automatic optimization of hazard factor selection, model construction, and parameter adjustment for flood risk assessment tasks, thereby improving efficiency, objectivity, and reproducibility in model development."

6. Line 95 - The authors have not provided any evidence to support this statement.

7. Line 96 - The statement is unsupported.

Response: Thank you for your valuable comments. We agree that the original statements in Lines 95–96 lacked sufficient supporting evidence. In the revised manuscript, we have restructured this section to improve clarity and provide appropriate citations. Specifically, we now cite Guo et al. (2022c) to support the effectiveness of AutoML in flood hazard prediction, and Hutter et al. (2019) and He et al. (2021a) to support its methodological advantages.

The revised text now reads:

"In the field of flood risk assessment, AutoML has been preliminarily demonstrated to perform well in flood hazard prediction (Guo et al., 2022). As an efficient 'black-box' modeling approach, AutoML provides strong support for flood risk modeling through automated feature selection, model training, and parameter optimization (Hutter et al., 2019; He et al., 2021). "

We believe this revision strengthens the foundation for the subsequent discussion on the methodological limitations of AutoML and the motivation for incorporating MCDA.

8. Line 99 -Which complex decision making problems are the authors referring to?

Response: Thank you for your insightful comment. We agree that the original sentence did not sufficiently clarify what types of complex decision-making problems are involved. In the revised manuscript, we have elaborated on this point by explaining that flood risk assessment in urban agglomerations involves multiple natural and socioeconomic indicators—such as rainfall, topography, land use, drainage, and population density — that originate from heterogeneous, often multi-source datasets and differ in type. These indicators frequently interact in non-linear and uncertain ways, making it difficult to evaluate their relative contributions to overall flood risk.

To address these challenges, we now clarify that multicriteria decision analysis (MCDA) provides a structured framework for integrating such diverse indicators into a unified evaluation system by constructing weighting schemes that align the results with real-world conditions and expert knowledge (Fernández and Lutz, 2010). Furthermore, in cases where certain data are missing or difficult to quantify, MCDA allows for the incorporation of expert judgment through tools such as scoring systems and pairwise comparison matrices, thus improving the robustness and practical applicability of the assessment (Hites et al., 2006).

The revised passage now reads:

In urban agglomerations, flood risk assessment is a highly complex task involving diverse natural and socioeconomic factors derived from heterogeneous and often multi-source datasets (Wang et al., 2023). These factors—such as rainfall, topography, land use, drainage, and population density—differ in type and often interact in non-linear and uncertain ways (Shuster

et al., 2005; Zhang et al., 2017; Wang et al., 2018). Under such complex circumstances, AutoML struggles to systematically evaluate the multi-dimensional indicators of flood risk. To address this limitation, this study introduces a multicriteria decision analysis (MCDA) approach to quantify the importance of various indicators within the evaluation framework (Pham et al., 2021). MCDA facilitates the integration of such heterogeneous indicators into a unified evaluation framework by constructing structured weighting schemes, thereby aligning the assessment results more closely with real-world conditions and expert knowledge (Fernández and Lutz, 2010). In cases where data are limited or certain indicators are difficult to quantify, MCDA methods allow for the incorporation of expert judgment through scoring systems and pairwise comparison matrices, enhancing the practical applicability and robustness of the model (Hites et al., 2006).

9. Line 171 - Was the normalization done prior to or after splitting the data into training and test sets? Doing it prior to split can give incorrectly optimistic results due to data leakage. For example, if training data only had GDP up to $1000 per capita, then the normalized value of 1.0 will refer to that. In test data, if a value of $1500 is observed, it would now be input as a value > 1.0 using the normalization logic from the training set. However, by normalizing prior to the split, the normalization logic would have already "seen" the $1500 value and assigned 1.0 to it, hence the data has been leaked.

Response: Thank you for your valuable comment. We appreciate your observation and acknowledge that the original wording in the manuscript may have led to misunderstanding.

We would like to clarify that in this study, the normalization was conducted after splitting the dataset into training and testing subsets, which effectively avoids data leakage. Specifically, the minimum and maximum values used for normalization were computed only from the training set, and the same values were used to normalize both the training and test sets. As a result, the normalized values in the training set are guaranteed to lie within the range [0,1], while values in the test set may exceed this range if they fall outside the value range of the training data.

To ensure clarity and correctness, we have revised the relevant sentence in the manuscript as follows:

(2) Normalization of the numerical range can be achieved using a normalization process. In this study, the Min-Max normalization method is applied. Specifically, the minimum and maximum values of each feature are computed only from the training set, and both the training and test sets are then normalized using these training-derived parameters. This ensures that the normalized values in the training set are scaled to the range [0,1], while the values in the test set may exceed this range if they fall outside the training set's value distribution. The formula is as follows:

$$x' = \frac{x - x_{min}^{train}}{x_{max}^{train} - x_{min}^{train}}$$

10. Line 177 - What is the definition of a flooded point given that multiple historical floods are being considered?

Response: Thank you for your insightful comment. We agree that the definition of a "flooded point" should be clearly stated, especially when multiple historical flood events are involved. In the revised manuscript, we have clarified that:

A flooded point is defined as a location that lies within the inundation extent of at least one recorded flood event during the study period.

This definition reflects the cumulative spatial footprint of flood events over time. All flooded points were selected from validated inundated areas identified through flood traces in the Global Flood Database and the EM-DAT database, and further verified through satellite imagery, Google Earth, and historical flood records. This ensures that the flooded points used in model training accurately represent locations that were affected by at least one historical flood.

11. Figure 4 - How were the second-level factors selected? Surely, there are 10's-100's of other factors each of which could contribute to H-E-V-C, so what led to the specific choices to select this subset?

Response: Thank you for your valuable comment. We agree that each dimension of the H-E-V-R framework includes a wide range of potential indicators.

In this study, the selection of second-level indicators was based on the following considerations:

(1) Theoretical framework and literature support: The indicator system was developed with reference to studies based on the H-E-V-R framework or similar vulnerability–resilience analytical approaches. It was specifically adapted to the characteristics of the Yangtze River Delta urban agglomeration. The selected variables have been widely used in existing flood risk assessments to quantify meteorological hazards, exposure, vulnerability, and resilience, and are proven to be representative and practical (Gain et al., 2015; Criado et al., 2019; Liu et al., 2019; Hsiao et al., 2021; Bin et al., 2023).

(2) Data availability and spatial applicability: All indicators used in this study are based on publicly available datasets with broad spatial coverage. Their resolution is appropriate for regional-scale analysis, particularly at the urban agglomeration level. This ensures consistency and comparability across the entire study area.

(3) Representativeness and reduction of redundancy: For each dimension, we prioritized key

indicators that reflect distinct aspects while avoiding strong correlations or overlapping meanings among variables. For example, in the exposure dimension, flood-related damage is typically correlated with the concentration of people, economic activity, infrastructure, and other assets in affected areas. Therefore, we selected indicators such as land area, population density, GDP density, and building density to capture different perspectives on the degree of exposure of natural and social systems to flood hazards.

(4) Modeling feasibility and efficiency: While ensuring comprehensive coverage of key factors, we also aimed to maintain interpretability and computational efficiency in the machine learning models. This avoids potential overfitting caused by excessive input variables. As a result, a representative and empirically grounded subset of indicators was finalized for use in this study.

Importantly, we would like to note that these principles and the rationale for indicator selection were already addressed in the previous revision of Section 2.4 ("Establishment of a flood risk assessment indicator system"). In that section, we clearly outlined the indicator selection process for each of the four dimensions (hazard, exposure, vulnerability, and resilience), explaining their relevance, definitions, and interpretation within the flood risk assessment framework. This earlier addition reflects our attention to scientific transparency and methodological justification.

12. Line 300 - How is the balance of Precision, Recall, and F1-score quantified?

Response: Thank you for your valuable comment. We have revised the relevant paragraph in the manuscript to clarify how the balance between Precision and Recall was quantified during model selection.

Specifically, the F1-score was used as the primary evaluation metric, as it provides a harmonic mean of Precision and Recall and is widely used to capture the trade-off between the two in binary classification tasks. For each set of hyperparameter combinations, k-fold cross-validation was performed within the training set, and the model with the highest average F1-score across all folds was selected as the optimal configuration. Precision and Recall were also reported independently to support interpretability, but F1-score served as the key criterion for quantifying the overall balance.

This clarification has been added to the revised manuscript (Section 2.5.2).

13. Line 419 - How was this 5-fold cross-validation performed, and how was it different than the k-fold used for hyperparameter tuning?

Response: Thank you for this important comment. We appreciate the opportunity to clarify the two distinct uses of cross-validation in our study.

In response to your suggestion, we have added a clarification at the end of Section 2.5.2 Model

training and hyperparameter optimization, explaining that 5-fold cross-validation was applied at two distinct stages:

(1) During hyperparameter optimization, 5-fold cross-validation was used within the training set only, to evaluate and select the best-performing hyperparameter combination.
(2) Following final model selection, an independent 5-fold cross-validation was applied to the entire dataset to evaluate the generalization performance of the model and identify potential overfitting.

The data splits used in these two stages were entirely separate, and no data leakage occurred. This clarification has been added to the revised manuscript to improve the transparency and reproducibility of the experimental procedure.

14. Line 485 - Did the authors populate the judgement matrix? How were the values selected?

Response: Thank you for your insightful question. We have revised Section 3.1.3 of the manuscript to clarify how the judgment matrices were constructed and how the values were determined.

Specifically, the judgment matrices were populated by the authors using a hybrid approach. For the hazard indicators, the pairwise comparisons were informed by feature importance scores obtained through the AutoML model. For the exposure, vulnerability, and emergency resilience indicators, the relative weights were derived based on a combination of expert judgment, a review of relevant literature, and the socio-environmental characteristics of the YRDUA. The Saaty 1–9 scale was applied to express the relative importance of each pair of indicators.

All judgment matrices underwent consistency checks, and the calculated consistency ratio (CR) was 0.0058, which is well below the 0.1 threshold, indicating that the matrices were logically consistent. These clarifications have been incorporated into the revised manuscript to enhance transparency and reproducibility.

15. Section 3.1.2 - Were only the hazard second-level factors used as features for the AutoML method, or were all 19 factors used, as only the importance of hazard factors is presented?

Response: Thank you for your important comment. We appreciate the opportunity to clarify the structure and feature input strategy of the AutoML model.

In this study, only the six second-level indicators under the hazard dimension were used as input features for the AutoML model. This is because flood hazard—defined as the physical likelihood of flooding—is primarily determined by natural environmental factors such as precipitation, elevation, slope, and drainage density. The AutoML algorithm was used to build a binary classification model (flooded vs. non-flooded points), which is a task it performs particularly well due to its strengths in automatic feature selection, high computational

efficiency, and predictive accuracy.

However, AutoML, as a black-box model, lacks the interpretability required for evaluating social or systemic dimensions such as exposure, vulnerability, and resilience. It cannot provide structured weights across multiple dimensions of flood risk. To address this, we incorporated the Analytic Hierarchy Process (AHP), a widely used multi-criteria decision-making method in flood risk assessment, to assign weights to these additional dimensions based on expert judgment and literature review.

The judgment matrix for the hazard indicators was constructed based on the feature importance scores output by the AutoML model. In contrast, the matrices for the exposure, vulnerability, and resilience dimensions were constructed using the 1–9 Saaty scale, informed by domain experts and relevant literature.

To ensure transparency, we have explicitly clarified this distinction in the revised manuscript in two places:

(a) In the final paragraph of the Introduction, we further clarified the overall model structure:

"This study develops a flood risk assessment model for the YRDUA by analyzing the factors influencing flood risk and integrating AutoML and AHP methods. In this model, AutoML is employed to construct the flood hazard sub-model, using indicators that represent natural environmental drivers as input features. The hazard is modeled as a binary classification problem (i.e., whether flooding occurs), and the resulting feature importance rankings provide an objective basis for subsequent indicator weighting. Nevertheless, as a data-driven approach, AutoML alone cannot structurally interpret the relative influence of social and systemic factors within a multi-dimensional flood risk assessment framework. Therefore, this study incorporates the AHP to calculate the weights of flood exposure, vulnerability, and resilience in the YRDUA, based on expert knowledge and existing literature. A regional flood risk zoning map is then generated. A comparative analysis with observed inundation points data shows a strong spatial alignment between the distribution of flooded points and the high to medium-high risk zones, highlighting the reliability and applicability of the proposed model. The remainder of this paper is structured as follows: Section 2 describes the study area, data sources, and methodology; Section 3 presents the results and analysis; Section 4 discusses the findings and their implications; and Section 5 concludes the study with key insights and recommendations."

(b) At the beginning of Section 3.1.2, we added the following explanation:

"In this study, the AutoML model was used specifically to assess flood hazard, which represents the physical likelihood of flood occurrence and is directly driven by environmental factors such as rainfall, topography, and drainage characteristics. Therefore, only the six second-level indicators under the hazard dimension were used as input features in the AutoML model. This

approach allowed us to focus the model on identifying the key natural drivers of flooding, while the other dimensions—exposure, vulnerability, and resilience—were later incorporated via the AHP method for comprehensive flood risk evaluation."

We hope this explanation and revision adequately address your concern.

16. Section 3.2 - Some of the factors like population and building density are expected to consistently increase and have likely done so between 1990 to 2010. What were the ranges of these factors during model training vs when using the model for calculating spatio-temporal variation? A table showing typical values both during training and later implementation will be helpful, and e.g., could be included in Table 1.

Response: Thank you for this thoughtful question. We believe this comment raises an important point regarding the temporal evolution of socio-economic indicators such as population density (DPOP) and building density (DBUI), and their implications for model training and application. We appreciate the opportunity to clarify how these variables were used in our study.

We would like to clarify that in this study, only the six second-level indicators under the flood hazard dimension—namely Average Annual Precipitation (PREC), Annual Cumulative Heavy Rainfall Duration (DURA), Digital Elevation Model (DEM), Slope (SLOPE), Drainage Density (DD), and NDVI—were used as input features for AutoML model training. These indicators are driven by natural environmental conditions and were used to train a binary classification model to predict flood occurrence (flooded vs. non-flooded points).

In contrast, other indicators such as population density, GDP density, and building density, which are known to evolve over time, were not included in the AutoML training process. Instead, they were integrated into the comprehensive flood risk assessment using the Analytic Hierarchy Process (AHP). This method enabled us to incorporate both expert judgment and literature references to assign weights to all second-level indicators across the three dimensions: exposure, vulnerability, and resilience.

This design reflects our overall methodological strategy of combining data-driven modeling (AutoML) with expert-informed decision analysis (AHP), enabling both predictive accuracy and model interpretability. Importantly, the AutoML component, which could be affected by variable shifts over time, was isolated from these time-sensitive socio-economic features, ensuring the model's temporal consistency and preventing leakage or extrapolation issues.

To further clarify this distinction, we have added an explanatory paragraph at the beginning of Section 3.1.2 of the revised manuscript.

We hope this explanation adequately addresses your concern.

[revised manuscript text omitted]